# Determinants of MDA impact and designing MDAs towards malaria elimination

Bo Gao[1], Sompob Saralamba[2], Yoel Lubell[1,2], Lisa J White[1,2], Arjen M Dondorp[1,2], Ricardo Aguas[1,2]*

[1]Centre for Tropical Medicine and Global Health, Nuffield Department of Medicine, University of Oxford, Oxford, United Kingdom; [2]Mahidol-Oxford Tropical Medicine Research Unit, Faculty of Tropical Medicine, Mahidol University, Bangkok, Thailand

**Abstract** Malaria remains at the forefront of scientific research and global political and funding agendas. Malaria models have consistently oversimplified how mass interventions are implemented. Here, we present an individual based, spatially explicit model of *P. falciparum* malaria transmission that includes all the programmatic implementation details of mass drug administration (MDA) campaigns. We uncover how the impact of MDA campaigns is determined by the interaction between implementation logistics, patterns of human mobility and how transmission risk is distributed over space. Our results indicate that malaria elimination is only realistically achievable in settings with very low prevalence and can be hindered by spatial heterogeneities in risk. In highly mobile populations, accelerating MDA implementation increases likelihood of elimination; if populations are more static, deploying less teams would be cost optimal. We conclude that mass drug interventions can be an invaluable tool towards malaria elimination in low endemicity areas, specifically when paired with effective vector control.

**\*For correspondence:**
ricardo@tropmedres.ac

**Competing interests:** The authors declare that no competing interests exist.

## Introduction

In Southeast Asia, and particularly the Greater Mekong Sub-region (GMS), *Plasmodium falciparum* transmission has decreased substantially over the last two decades (*World Health Organization, 2011*; *World Health Organization, 2010b*), setting the stage for pre-elimination scenarios, with all GMS countries committing to ambitious elimination timelines (*World Health Organization, 2013a*). Alignment of global funding bodies' goodwill with sound national malaria control programmes is crucial for elimination timelines to be met (*World Health Organization, 2013c*; *Alonso and Tanner, 2013*), but spreading artemisinin resistance creates a race against time before malaria becomes untreatable with currently available drugs (*World Health Organization, 2010a*; *World Health Organization, 2013b*).

Vector control and early diagnosis followed by effective antimalarial treatment have been the mainstay of malaria control programmes, but modelling based projections indicate these approaches alone are unlikely to achieve *P. falciparum* malaria elimination before failing drug efficacy becomes an issue. Elimination will require more intensive measures to clear the infectious reservoir in asymptomatic populations, especially in the GMS where existing vector bionomics make vector control particularly challenging. The most abundant vector species in the GMS are exophilic (mainly bite outdoors), do not preferentially bite humans, and can bite quite early in the evening (*Sinka et al., 2011*), rendering typical vector control measures such as insecticide treated nets (ITNs) and indoor residual spraying (IRS) sub-optimal.

Population wide interventions, including mass drug administration (MDA), are under consideration to clear the infectious reservoir in asymptomatic populations and potentially hasten progress

toward elimination (*Poirot et al., 2013*; *World Health Organization, 2015a*). The proportion of the target population receiving these interventions ('coverage') is believed to determine their success (*World Health Organization, 2013c*; *Slater et al., 2015*; *Okell, 2015*; *Stuckey et al., 2016*). This success can be considered at two spatial levels: global or local. Whilst malaria elimination campaigns have been carried out successfully in some countries or locally in specific regions (*Snow et al., 2013*; *John et al., 2009*; *World Health Organization, 2015b*), reintroductions of malaria from surrounding endemic areas are a constant threat (*Cohen et al., 2012*; *Galappaththy et al., 2013*). The importance of mobile populations as a source of malaria transmission in the GMS has been emphasized in recent years (*Pindolia et al., 2012*; *Prosper et al., 2012*; *Pindolia et al., 2014*; *Smith and Whittaker, 2014*; *Edwards et al., 2015*; *Guyant et al., 2015*). Prompt treatment of new clinical cases through village malaria worker (VMW) or village health worker (VHW) networks has proven to be an effective case management strategy (*Maude et al., 2014*; *Rutta et al., 2012*) and would be an essential barrier against malaria reintroduction.

We argue that the way in which mass interventions are deployed is what determines their success likelihood. The most efficient and effective roll-outs are laid on a solid community engagement foundation, thus ensuring subsequent adherence and coverage, while preventing malaria reintroduction from adjacent areas. Here, we model target areas as a collection of discrete villages (unit of intervention) and define coverage as the proportion of individuals receiving the intervention within a village and also as proportion of villages receiving the intervention within an area. Critically, we also simulate the minutia of mass intervention roll outs, with all its relevant deployment logistics, thus assigning coverage a temporal dimension which measures the time it takes for all target villages to receive the intervention.

Conducting enough clinical trials to understand the interaction between all variables at play during a mass drug administration, as well as their individual and combined contribution to the expected outcome, is prohibitive. Hence, we turn to computational modelling to explore the relationships between logistical aspects of MDA implementation and demographic aspects such as human population mobility in diverse epidemiological settings (characterized by prevalence, seasonality patterns and heterogeneity in mosquito densities across space). Our focus in on how the predicted impact of mass intervention strategies on malaria transmission changes when these logistical intricacies are taken into consideration, and its implication for the likelihood of *P. falciparum* elimination. Our research questions are threefold: 1) what is the relevance of logistical implementation details to the outcome of mass interventions? 2) How fast does target coverage need to be reached for the strategy to be successful? 3) What are the key modulators of malaria elimination likelihood in a short timeframe? To offer strategic guidance to national malaria control programmes we also need to understand how the answers to these questions hinge on key features of malaria transmission in specific areas such as artemisinin resistance levels, population mobility networks, transmission heterogeneity over space, and seasonality patterns.

## Model description

We developed a modular simulation platform that is customizable to any malaria transmission setting to provide realistic outcome predictions for local and global level interventions. The modules are the building blocks of an individual based, discrete time, spatially explicit, stochastic model, with explicit mosquito population dynamics and human population movements. We thus have villages with different mosquito densities connected by a human flow network, on which different interventions are deployed at different times (*Figure 1*). One particular innovation compared to previous published work (*Gatton and Cheng, 2010*; *Okell et al., 2011*; *Maude et al., 2012*; *Gerardin et al., 2015*; *Nikolov et al., 2016*) is the inclusion of very detailed logistical processes related to intervention deployment in the field.

Whilst previously published models are extremely good at representing the biological processes underlying malaria transmission, some even making very realistic assumptions on how coverage increases over time (*Nikolov et al., 2016*), they fail to explicitly model how these interventions are carried out in the field. In practice, teams of workers visit villages sequentially one by one, usually spending 4–7 days to deploy one MDA round in each village, which is quite different from having an unlimited number of teams slowly treating everyone in the target population until a certain coverage is reached. The number of implementation teams is then an input parameter in our simulation platform and each team behaves as an agent. They remain in each village for a fixed period of time

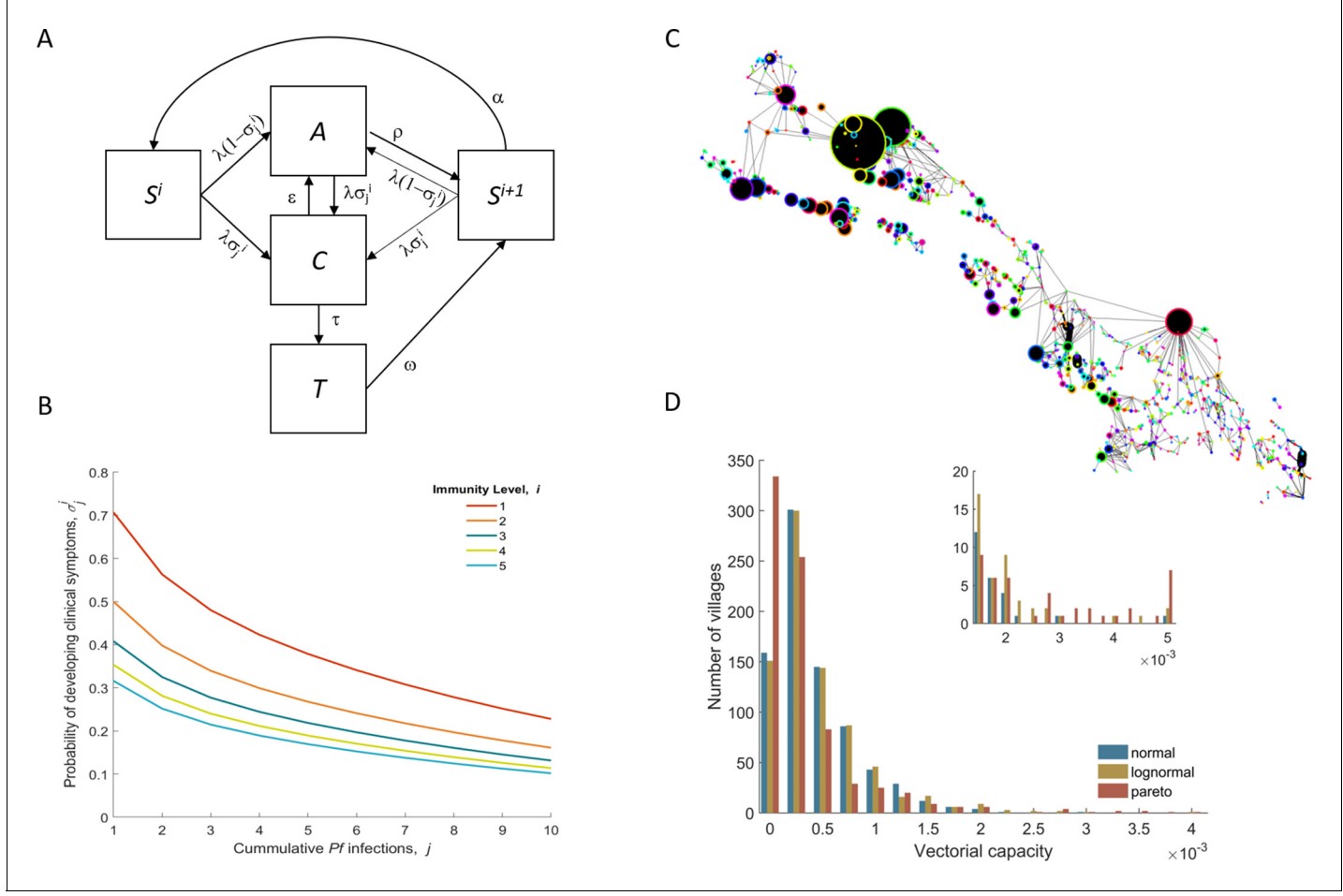

**Figure 1.** Model structure illustration. (**A**) Flow diagram representing the natural history of *Plasmodium falciparum* infections in human populations. Uninfected individuals (S) can be infected at rate $\lambda$, with the probability of developing clinical symptoms ($\sigma$) depending on their immunity level $i$ and number of lifetime infections $j$. Clinical infections (C) can be detected and subsequently treated at rate $\tau$ to a treated state (T), or naturally subside into an asymptomatic parasite carrier stage (A) at rate $\varepsilon$. Treated individuals lose their drug at rate $\omega$. After recovery or treatment, individuals become susceptible with an added level of clinical immunity ($S^{i+1}$). Clinical immunity level decays at rate $\alpha$. (**B**) Probability of developing clinical malaria depending on individual's history of infection (cumulative number of infections) for each immunity level considered here. (**C**) Village connectivity network. Geo-located villages appear as circles the size of which is proportional to the number of people living there. Edge width reflects individuals' probability of travel between connected villages. (**D**) Each village is assigned a specific mosquito density/vectorial capacity, with transmission heterogeneity over space characterized by three different distributions.

The online version of this article includes the following figure supplement(s) for figure 1:

**Figure supplement 1.** Synthetic population demographics.
**Figure supplement 2.** Model fit to immunological data.
**Figure supplement 3.** Calibrated relationship between EIR and Data.
**Figure supplement 4.** Calibrated relationship between prevalence and clinical malaria incidence.
**Figure supplement 5.** Clinical age profiles for different endemic levels.
**Figure supplement 6.** MDA effect size for different values of starting prevalence.

(*process = 4 days*), assumed to be the time needed to complete one round of MDA per village, and move to the next village according to a gravity model. Given a total number of villages *V*, and a processing time *process*, the total number of days taken from start of first MDA round in the first village to end of first MDA round in the final village (D) depends on the number of implementing teams (Teams): D = *V*\*process/Teams.

# Results

Initially, we simulated thousands of parameter sets that explore how a wide range of key transmission parameters (e.g. mean initial parasite prevalence, proportion of artemisinin resistant parasites) and logistical constraints – *Table 1* – modulate the expected outcome of MDA campaigns. *Figure 2* illustrates the sensitivity of the predicted proportional decrease in prevalence over 5 years to each parameter. Clearly, the number of MDA campaigns and the initial mean prevalence across all villages are critical covariates when predicting MDA outcome. The distributions characterizing how malaria risk is distributed over space also seem quite important. Artemisinin resistance spread is very sensitive to those same covariates as well as to the number of intervention teams and the intensity of human population mobility (*Figure 2—figure supplements 4* and *5*).

We found that there is an intricate relationship between the optimal timing of MDA campaign start, its implementation logistics, and malaria seasonality patterns. Deploying a higher number of MDA teams will yield a higher likelihood of reaching malaria elimination within 2 years, only when population mobility is high – *Figure 3*. Using a smaller number of intervention teams is predicted to be advantageous in a population of lower mobility, especially when there is only one annual transmission peak and when the MDA start is delayed to day 60 (instead of the default start at the beginning of the calendar year). In settings with 2 malaria seasons per year, the first transmission peak occurs earlier in the year, making the faster 400 team implementation a better option in general. The only exceptions are very static populations in which two MDA campaigns are deployed. We should note that overall, a slower implementation is preferable, especially for the single peak seasonal profiles (*Figure 3—figure supplement 1*). For the two peak scenarios, a higher number of teams would be beneficial as prevalence increases from 1%. When addressing how to maximize the chances of reaching elimination within a short time span by implementing an MDA strategy, we found that different transmission heterogeneity distributions (depicted in *Figure 1D*) incur quite different prospects – *Figure 4* (Normal), *Figure 4—figure supplement 1* (Log-Normal), and *Figure 4—figure supplement 2* (Pareto). The likelihood of reaching elimination is strikingly different when comparing the Pareto distribution (most skewed) with the other two (Normal and Log-Normal distributions), except when an extremely efficient vector control program is carried out for a couple of years. Coupling vector control with an MDA campaign vastly improves the chances for elimination across the low prevalence settings explored here, and the longer vector control can be sustained the more likely elimination becomes. Once again, there seems to be a correlation between population mobility and number of MDA teams, with faster MDA implementations being preferred when the human population is more mobile – *Figure 4*.

**Table 1.** Factors explored by the model and their respective sets of values.

| Factor | Meaning | Values |
|---|---|---|
| Teams | Number of teams performing prevalence surveys and distributing ACTs simultaneously. Translated into coverage speed in square brackets, that is, the number of days it takes to perform one MDA round in a region of 1000 villages. | 15 [267]<br>25 [160]<br>50 [80]<br>100 [40]<br>200 [20]<br>400 [10] |
| Prevalence | Mean initial malaria u-PCR prevalence across all villages. | 0.5, 0.1, 0.15, 0.2, 0.25, 0.3 |
| Resistance | Mean initial proportion of parasites which are artemisinin resistant. | 0.1, 0.2, 0.3 |
| Distribution | Transmission heterogeneity distribution, underlying spatial heterogeneity of malaria transmission, manifested by differential mosquito density distributions. | Gaussian, Log-normal, Pareto |
| Mobility | Describes the intensity of population movement in the general population. Indicates the per person average daily probability of moving to places other than their home village for short-term visits (see Mobility section in the Simulation Protocol). that is a population of individuals whose short-term movement occurs on average 25 times per year (25/365) has a relatively static population with low mobility. | 5/365<br>25/365<br>125/365<br>250/365 |
| Campaigns | The number of MDA Campaigns | 1, 2 |
| Peaks | The number of annual seasonal peaks | 1, 2 |

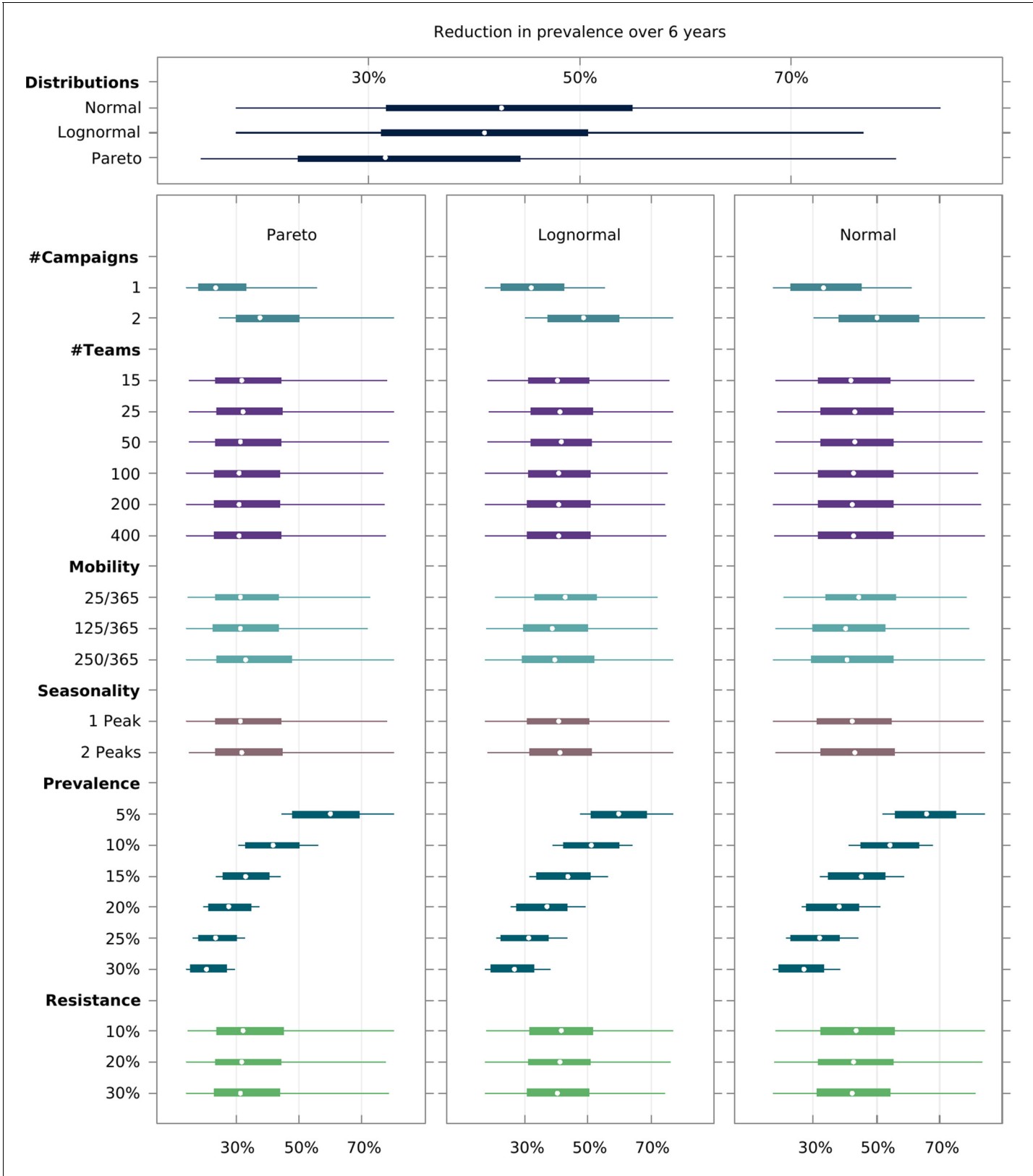

**Figure 2.** Multivariate sensitivity analysis of the predicted intervention impact on malaria prevalence and artemisinin resistance. The box plots show the median and interquartile ranges of the proportional reduction in malaria prevalence for all simulated parameter sets. Each parameter set consists of a combination of initial mean prevalence (*Prevalence*), initial proportion of artemisinin resistance parasites (*Resistance*), population mobility (*Mobility*),

*Figure 2 continued on next page*

*Figure 2 continued*

number of teams deployed in the field (# *Teams*), number of MDA campaigns (# *Campaigns*), and number of transmission peaks per year (*Seasonality*). An overall mean and interquartile range for the effect of transmission heterogeneity, independent of any other parameter, is displayed on the top panel. The reduction in prevalence is evaluated as the proportional difference in the integral in prevalence in the 5 years following MDA relative to the 5 years preceding MDA.

The online version of this article includes the following figure supplement(s) for figure 2:

**Figure supplement 1.** Multivariate model sensitivity analysis independent of transmission heterogeneity.
**Figure supplement 2.** Interplay between population mobility and transmission heterogeneity.
**Figure supplement 3.** Interplay between logistics and human population topologies.
**Figure supplement 4.** Factors influencing the spread of artemisinin resistance.
**Figure supplement 5.** Population movement and mosquito distributions determine artemisinin resistance spread.

Finally, we explored the value of targeting the top 10 or 20% of villages (sorted by vectorial capacity) and compared its predicted outcome with a full MDA campaign – *Figure 5*. We confirm that the log-normal and gaussian distributions explored here produce the same elimination likelihood profiles. For very efficacious vector control strategies, the vectorial capacity in the transmission foci will be greatly reduced, causing a greater drop in mean vectorial capacity across all villages in the more skewed distribution (where most villages have negligible numbers or no mosquitoes), compared to the others. This causes the likelihood of elimination in settings with a Pareto distributed risk of infection to be greater on the long-term under those circumstances. Once again, sustaining the vector control for longer, greatly improves the expected outcome (*Figure 5—figure supplement 1*).

## Discussion

A series of key interacting features of the transmission-intervention system emerge when intricate logistics are incorporated in spatial-temporal transmission dynamics. Mapping MDA campaign expected outcomes to a specific malaria endemic setting is a complex multivariate problem. Here, we elucidate the way in which the most critical interactions determine MDA success:

- Operational strategy design. Mass intervention strategies rely on a detailed protocol defining the proportion of villages targeted, the target population in each village reached (usually termed target coverage), and the number of intervention teams deployed (determining the speed with which all villages are covered). Unsurprisingly, the chosen number of MDA campaigns is the most significant intervention outcome determinant (*Figure 2*). This is intuitive in a scenario where treatment failure due to drug resistance is not a serious issue. Assuming 80% of the target population receives each MDA round, and independent coverage between rounds (meaning the likelihood that someone adheres to round 3 for example is independent of their uptake in rounds 1 and 2), by increasing the number of MDA rounds, we are decreasing the proportion of the population not treated with at least one round of ACT. Even if adherence and compliance are correlated, increasing the number of rounds would assure individuals that received treatment would be less likely to become infectious, or be infectious for long, if infected via untreated individuals. Indeed, additional MDA rounds provide a powerful tool to disrupt any resurgence in transmission following the typical 3 round MDA campaigns. The likelihood of elimination being achieved is substantially higher for 2 campaigns of 3 MDA rounds compared to 1 campaign (*Figure 3*).

- Transmission heterogeneity, described by different mosquito density distributions over space, and initial mean parasite prevalence in the human population also have a clear impact on the predicted reduction in prevalence with MDA (*Figure 2—figure supplements 1* and *2*). Of note, MDA strategies on their own are not predicted to achieve elimination unless mean malaria prevalence is at very low levels (under 3%) – *Figure 2* and *Figure 3—figure supplement 1*. Coverage speed, increased with a higher number of intervention teams, is programmatically beneficial only when elimination is achievable (prevalence <3%) and in scenarios where MDA is deployed in well-connected populations (consisting of individuals with high mobility) – *Figure 3*. Counterintuitively, in all other scenarios, a slower MDA implementation is optimal (*Figure 2—figure supplement 3*, *Figure 3—figure supplement 1*). When the number of MDA teams is highest, all villages receive the first round within a couple of weeks, meaning a very large proportion of the population will be under treatment simultaneously. Whilst that

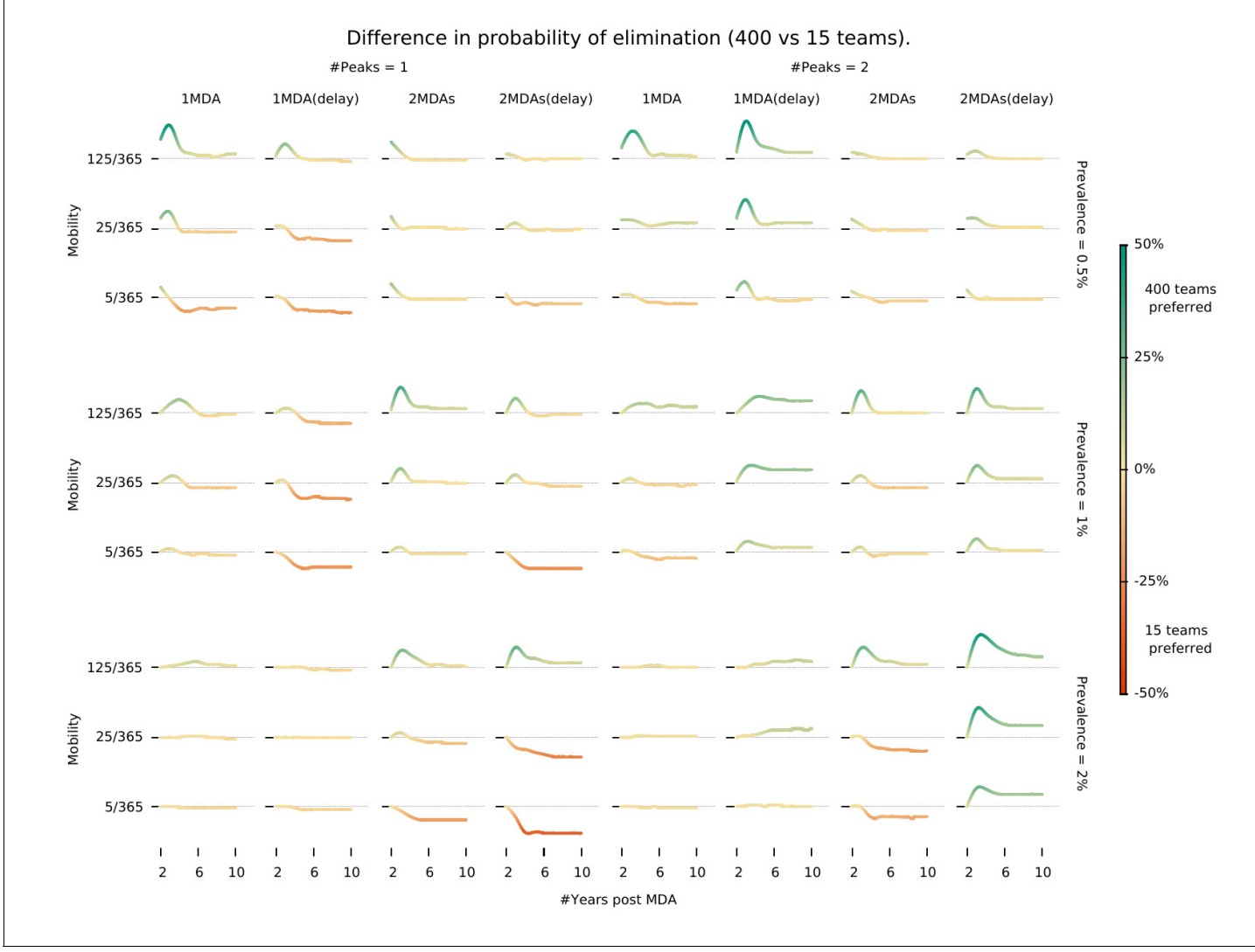

**Figure 3.** Intervention implementation speed in different epidemiological contexts. Demonstrates under what conditions using 400 implementation teams is preferable over 15 teams when deploying an MDA campaign. We investigate different epidemiological contexts, characterized by different prevalence and seasonality profiles can be accounted for in deciding the appropriate campaign start day (delay) when maximizing the chances for malaria elimination. 'MDA (delay)' means MDA start is delayed to day 60 instead of the default start at the beginning of the calendar year. '#Peaks' indicates the number of transmission peaks per year. The left four columns have 1 peak whereas the right four columns have 2 peaks.

The online version of this article includes the following figure supplement(s) for figure 3:

**Figure supplement 1.** Elimination likelihood over time in different settings.

**Figure supplement 2.** Relationship between elimination probability and MDA campaign village sequencing.

**Figure supplement 3.** Relationship between elimination probability and the relative infectivity of asymptomatic infections.

translates into the largest possible increase in the likelihood of elimination, any resistant infections will have a large selective advantage at that point, causing resistance to spread (*Figure 2—figure supplement 5*). Interestingly, this effect is less pronounced in populations with high mobility, due to a dilution effect described below. A slower deployment of MDA campaigns is then generally preferable in low mobility populations (here defined as settings where individuals spend on average less than 25 nights per year somewhere other than their home) due to a slower buildup of resistant infections (*Figure 4*, *Figure 2—figure supplement 5*).

- Interestingly, prioritizing the very first villages to receive MDA will have very little effect on the MDA campaign's success (*Figure 3—figure supplement 2*). This is solely due to the spatial dispersal of transmission foci. In a setting where transmission foci are scattered over space (as simulated here), not much can be gained by prioritizing high incidence villages to receive

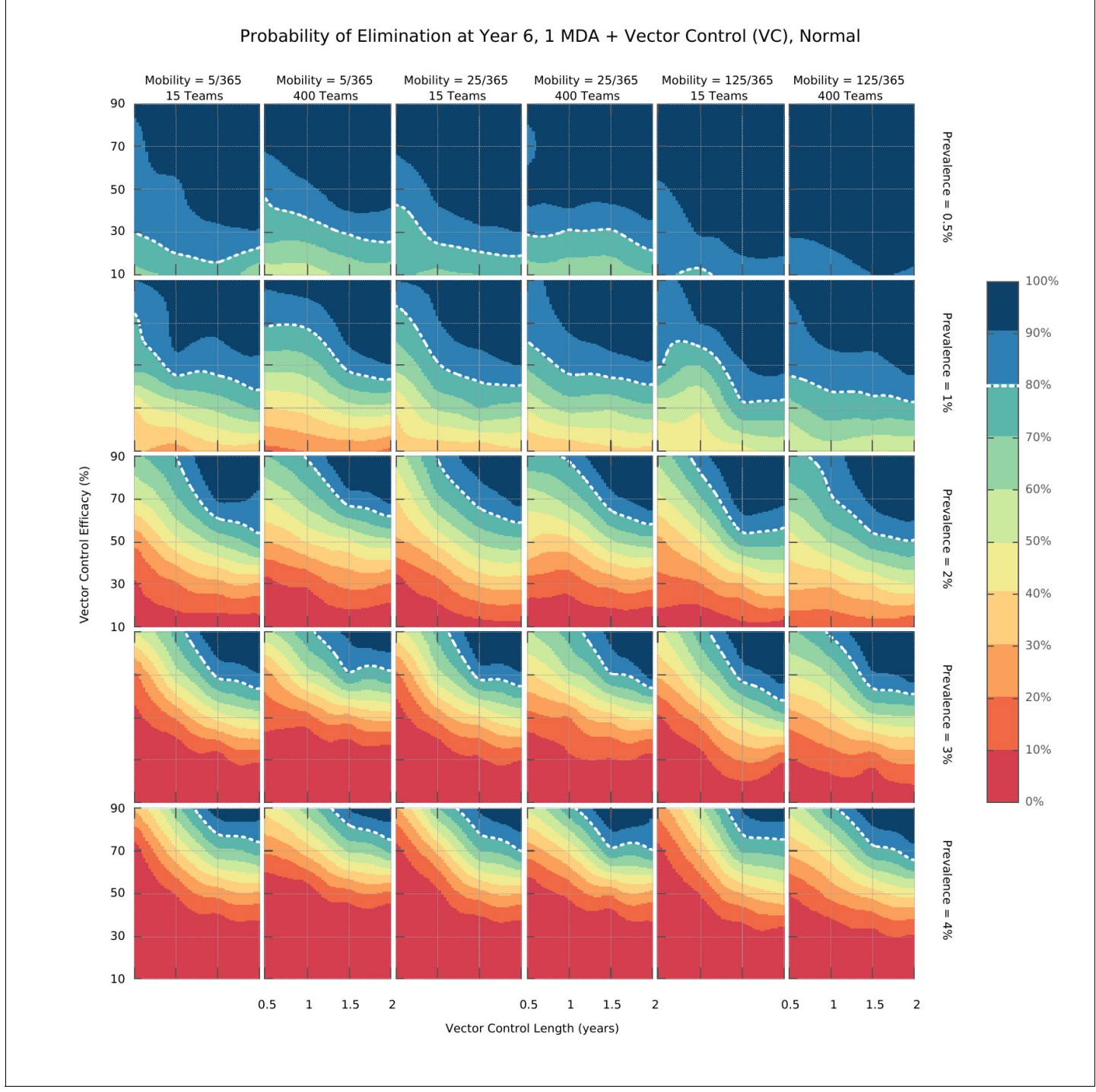

**Figure 4.** Elimination probability surfaces (Normal distribution of transmission risk over space). These surface plots show the proportion of simulations (out of 100) in which elimination was achieved within a 5 year time horizon. Hypothetical interventions that decrease the vectorial capacity by a proportion given in the y-axis are maintained for a period of time defined in the x-axis. These transmission blocking interventions are layered on top of a global MDA campaign consisting of 3 ACT rounds in all villages and including widespread village malaria workers. Different panels give different combinations of mean initial malaria prevalence, human population mobility and MDA implementation speed. The white dashed line represents the 80% likelihood of elimination contour line.

The online version of this article includes the following figure supplement(s) for figure 4:

**Figure supplement 1.** Integrated control elimination surfaces for the Log-normal distribution of transmission risk over space.

**Figure supplement 2.** Integrated control elimination surfaces for the Pareto distribution of transmission risk over space.

**Figure supplement 3.** Mean reduction in prevalence in intervention strategies containing either MDA + VC or VC alone.

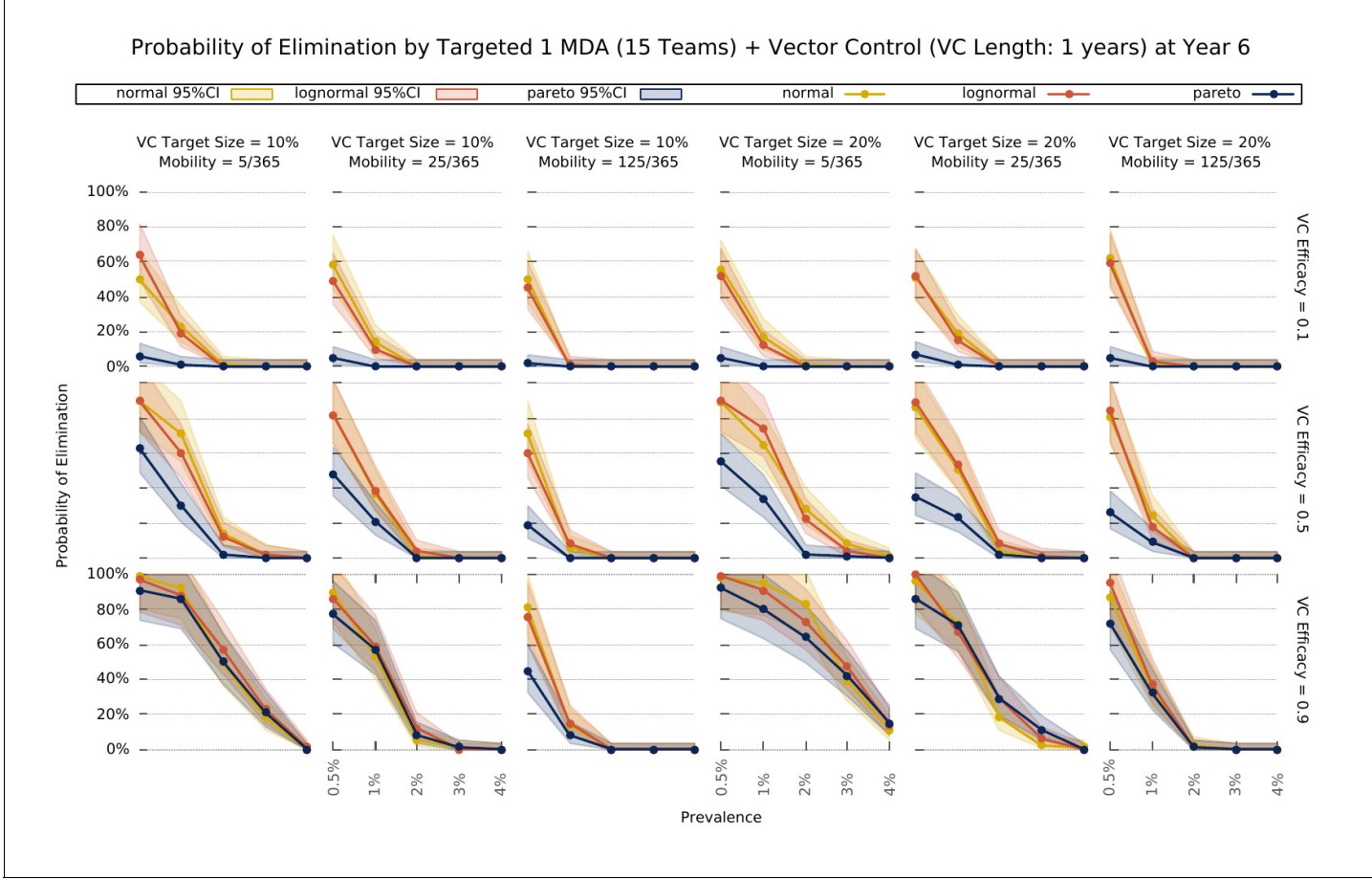

**Figure 5.** Elimination probability with a targeted approach. Illustrates the likelihood of elimination within 5 years of an elimination strategy consisting of 3 MDA rounds and a vector control strategy sustained over 1 year. We compare elimination prospects across different prevalence levels, human population mobility, and transmission heterogeneity over space. Vector control (VC) efficacy refers to the coefficient by which vectorial capacity is reduced for the duration of the intervention. The vector control target sizes refer to the quantile of villages, sorted by descending vectorial capacity, targeted by the intervention.

The online version of this article includes the following figure supplement(s) for figure 5:

**Figure supplement 1.** Elimination probability with a targeted approach.

MDA fist, since once the intervention teams leave those villages, transmission will be re-seeded from lower incidence neighboring villages or forest areas. In fact, if there is only one seasonal peak and only one MDA campaign is carried out, staring in lower incidence villages is predicted to incur a statistically non-significant benefit. Note that in areas where foci are clustered in a small area, a targeted MDA approach would be preferable over a global MDA, with villages outside the foci area not receiving MDA.

- Transmission topology. This is defined by the magnitude of transmission heterogeneity over space combined with the level of mixture between sub-populations through human movement. As mentioned above, spatial transmission heterogeneity has a dramatic effect on predicted outcomes, with more skewed distributions (suggesting most malaria infections occur in a few villages) presenting a challenge for control (*Figure 4—figure supplement 2*). We should note that *P. falciparum* transmission in the Greater Mekong Sub-region (GMS) has been reported to be spatially heterogeneous (*Gryseels et al., 2015*; *Cui et al., 2012*; *Erhart et al., 2005*). However, there is very little evidence as to the relative abundance of the main vector species and no quantification of densities exists at a large enough scale, thus we investigated a spectrum of population mobility patterns which at one extreme consists of a set of isolated transmission foci (thus low population movement), where infections in each village are almost exclusively locally acquired (mosquitoes infecting someone will have had acquired that infection from someone else living in the same village). As population mobility increases, these foci

become more and more connected, eventually merging with each other in the upper extreme of the spectrum, onto one large homogeneously mixed population (where mosquitoes infecting a person could have acquired that infection from anyone else).

- In low connectivity populations (consisting of individuals with low mobility), the likelihood of resurgence in villages where MDA achieved local elimination is low, since there are only very sporadic introductions of parasites, typically insufficient to reseed endogenous transmission. In these settings, implementation speed should be sacrificed, and a low number of intervention teams deployed, to minimize resistance spread. In populations with high population movement, speed of implementation becomes more important (*Figure 3*) due to the propensity for recently eliminated intervention units to be reseeded by its neighbors, leading to local resurgences.

- Seasonality. The optimal timing of MDA campaigns relative to malaria seasonal peaks have been theoretically investigated in *Brady et al. (2017)*. Critically, those models do not incorporate implementation logistics in a detailed manner, either simulating instantaneous MDA deployment in all villages, or having a synchronously increasing global coverage until a target coverage is reached. Here, we have explicit intervention teams that deploy sequential MDAs, one village at a time. This is much closer to reality in the field and creates an added dimension when comparing MDA start with malaria seasonal peaks. A very slow implementation that starts 3 months prior to the peak might only have reached a 50% coverage by the time transmission intensity hits the peak, whereas a very fast MDA deployment might start 1 month prior to the peak and end before it.

- When there is only one transmission peak during the year, a slower MDA implementation seems to be preferable (*Figure 3*), especially if the start of the MDA is set to start one month prior to the peak in vectorial capacity (not to be mistaken with the season malaria incidence peak which occurs later) instead of starting at the beginning of the year. It seems delaying the start of MDA campaigns improves the likelihood of malaria elimination compared to a start at the beginning of the year when a lower number of MDA teams is used (*Figure 3—figure supplement 1*). This is mostly due to how sensitive near instantaneous MDA campaigns are to the timing of the seasonal peak. In a single annual peak setting, where the incidence peak is at day 140, a near instantaneous MDA would end (all 3 rounds) 70 (for no delay) or 10 (with delay) days prior to the peak. Given the general cosine function simulated here, it seems a longer implementation lasting the whole duration of the high transmission season is optimal. For the two annual peaks scenario, a fast implementation of MDA (covering the whole of the first annual peak) seems preferable.

- Dilution. We uncovered an interesting trade-off between population mobility, transmission heterogeneity and number of MDA teams that results in unexpectedly high predictions for intervention impact in highly mobile populations. This is due to a diluting effect, rooted in the sharing of parasite pools between high transmission foci and very low transmission villages, which is particularly relevant in the post MDA rebound period. Granted high population mobility, after the parasite pool is greatly reduced through MDA, the few infectious mosquitoes in transmission foci are likely to bite migrants, which upon return to low transmission villages, are unlikely to transmit those infections onwards. Increased mobility decreases the proportion of endogenous infections in each village, which consequently increases the radial impact of local interventions, thus also generally contributing to the dilution of artemisinin resistant emerging infections (*Figure 2—figure supplement 5*). In a scenario where village A with an extremely low vectorial capacity is very close to a high transmission village B and there is intense population movement between villages, it is likely that infections in people living in village A are almost exclusively acquired when they visit village B. Thus, an MDA in village B will have a positive knock-on effect on the incidence of malaria in village A.

- Drug Resistance. We observe that increased transmission heterogeneity is detrimental to intervention success (*Figure 4—figure supplement 2*), through a mechanism whereby high transmission foci provide a niche for resistance spread which is promoted by drug pressure incurred through multiple MDA rounds (*Figure 4*, *Figure 2—figure supplement 4*). Interestingly, this effect is more pronounced for populations with the lowest mobility for all considered spatial heterogeneity distributions. This is due to the dilution effect mentioned above, causing artemisinin resistant parasites expanding in high transmission foci to be diluted across other villages, given a sufficiently high population movement. If the population is very static, however, resistant parasites can gradually outcompete sensitive ones in transmission foci receiving multiple MDA rounds. Drug resistance is also a key driver of why more heterogeneous topologies are predicted to have a lower MDA impact for low mobility populations (red lines in *Figure 2—figure supplement 5*). Thus, any concerns regarding the enhancement of artemisinin

resistance spread with MDA hinge on the transmission heterogeneity of the setting in which MDA is deployed. This entails serious consequences for the likelihood of malaria elimination with an MDA approach in populations where resistance is already established.

Although the elucidation of these intricate relationships is of great scientific interest, National Malaria Control Programmes (NMCPs) might find this exploration to be devoid of applicability, specifically those concerned with drug resistance issues. We have demonstrated that more MDA rounds translate into higher likelihoods of malaria elimination but also show how resistance is very likely to increase dramatically if elimination is not achieved. To explicitly inform policy decisions of NMCPs aiming to eliminate *P. falciparum* malaria in a short time frame, we provide insights into integrated strategies that combine vector control interventions with a minimal number of MDA rounds:

- Intervention layering and elimination prospects. While MDA strategies consisting of only 3 rounds of ACT are unlikely to interrupt transmission in all but very low prevalence (<2% all-age true prevalence) settings, the village malaria worker (VMW) network providing operation support to MDA campaigns does provide a great foundation for additional interventions to be more easily deployed. Combining a complete VMW network with 3 rounds of MDA at 80% coverage and imperfect vector control strategies, we predict malaria elimination can be reached, provided the initial parasite prevalence is sufficiently low (under ~2%) – *Figure 4*. Whilst vector control clearly provides a significant increase in the chances of reaching elimination, it would be unlikely to lead to elimination on its own as seen in *Figure 4—figure supplement 3*. More than a theoretical possibility, elimination, even when in the presence of artemisinin resistance, has been demonstrated to be possible (*Lwin et al., 2015*). This bistability phenomenon, first proposed for malaria a decade ago (*Aguas et al., 2008*) and since revisited (*Smith et al., 2013*), provides the theoretical foundation for the determination of the minimum intervention effort sustained over a defined period of time, after which all intervention measures can be relaxed and elimination is still reached (provided clinical case management remains effective). We thus explored the prospects for elimination with realistic intervention packages coupling vector control with MDAs using our simulation platform – *Figure 5*. We find that initial mean prevalence is a key determinant of success likelihood and determines the minimum effect size required from the vector control component, for elimination to be reached. We highlight how logistical constraints combine with human movement patterns to modulate the likelihood of an intervention strategy's success. Clearly, when population mobility is high, elimination becomes likely even with low vector control effect sizes, if MDA is done near instantaneously (400 intervention teams). A much higher vector control effect size would be required if MDA implementation is significantly slower. Conversely, if the human population is static, a slower MDA implementation would increase the likelihood of elimination for the same vector control effect sizes. This illustrates quite well the value of the missing information in simulation models that convert the highly complex logistics of a global MDA administration into an instantaneous process.
- Targeting interventions. Over the last few years, MDA interventions have moved towards a focal approach, where only a proportion of individuals within a village (*Eisele et al., 2015*), or a proportion of villages in the target area will receive treatment (*METF, 2016*), to ease drug resistance spread concerns, and to minimize the number of ACT doses given to uninfected individuals. The purpose is to implement MDAs in high transmission foci only, thus bringing mean prevalence across the whole population down very quickly, and then relying on good clinical case management and vector control interventions to eventually reach elimination. The logistical implementation is also simplified, and its associated costs minimized, since only a fraction of MDAs are performed. We explored the likelihood of reaching malaria elimination within 5 years by implementing a targeted elimination program combining 3 rounds of MDA with different vector control intensities and found that the duration and effect size of the vector control component greatly influences the prospects for elimination. Interestingly, increasing the target size (number of villages receiving the intervention), minimizes the differences across transmission heterogeneities. In fact, the most skewed distribution offers better elimination prospects for higher prevalence settings where intense vector control measures (VC efficacy = 0.9) are put in place. The nature of the distribution provides the basis for this effect. Given that most villages have a negligible vectorial capacity, transmission is sustained by a few high transmission foci. If those are targeted efficiently, you can expect to have a disproportionately higher disruption of transmission compared to settings where the distribution of mosquitoes over space is more homogeneous.

We have refrained from doing a cost-effectiveness analysis, since we do not have enough information on most unit costs, which are currently being assessed in different settings, and are likely to be quite variable across countries in the GMS. Any recommendation and cost-effectiveness analysis would have to be tailored to each specific country/area. We also have not extensively addressed synchronous migration patterns but have included long term and seasonal migration events in the simulation platform. The sensitivity of the model's predictions to these types of migration is much lower than to the general population's short-term mobility actions explored to great lengths throughout. This is, in all likelihood, a result of the low proportion of seasonal or long-term migrants in the overall population. In settings where migrants constitute a more considerable (>20%) fraction of the population, the predicted impact might vary. We also considered an uncorrelated uptake of ACT rounds during MDA, meaning there is no relationship between the likelihood of receiving a future ACT round and having received a previous one. This can be an issue in areas where religious and/or cultural beliefs cause individuals to refuse any and all drug treatments, but if that is the case, then it would manifest itself at the time of the prevalence survey, when blood would have to be drawn. In practice, we concede that we may be unable to deploy MDA in whole villages due to these constraints, but in the absence of data we refrain from making any assumptions. For simplicity, we have assumed that asymptomatic and clinical infections are equally infective to mosquitoes throughout. In reality, further empirical studies are greatly needed to better characterise this controversial quantity that bears critical consequences for malaria elimination prospects (*Aguas et al., 2018*). In *Figure 3—figure supplement 3* we demonstrate this critical nature, with malaria elimination becoming more amenable if asymptomatic infections are less infectious. To keep the mosquito population comparable to that in all other simulations shown here, we leverage the biting rate in simulations with lower infectivity of asymptomatic infection to obtain comparable prevalence levels. That means that for a given prevalence value, we took the mosquito density in previous simulations and calibrate the biting rate (proxy for effective number of human/mosquito contacts) needed to obtain the same prevalence at equilibrium when infectivity is lowered. Since the overall infectiousness of asymptomatic infections is decreased, the biting rate will have to increase to reach the same level of prevalence. Therefore, the infection pool will be sustained in a smaller population of mosquitoes which bite humans more frequently to make up for the decrease in human to mosquito infection efficiency. We can then conclude that the true catalyst of malaria elimination is the crash of the infectious population of mosquitoes after MDA, rather than a complete elimination of infections in the human population.

Here, we present a theoretical exploration of the potential impact of MDA strategies in different settings of the GMS, with special emphasis on the sensitivity of the predicted impact to logistical constraints, and transmission or population topologies. The ranges of parameters and distributions explored are meant to represent the current malaria situation in the GMS but need to be adjusted for application to specific areas/countries. In conclusion, we propose that mass drug interventions can be an invaluable tool towards malaria elimination in the right context. The model presented here predicts that an MDA's success likelihood is bounded by the initial malaria prevalence and we elucidate how those chances can be improved through tailoring of implementation logistics. Although MDA is being revisited by the global community, very little attention has been paid to implementation logistics, and there seems to be no protocol adjustment across settings with completely different seasonality and human mobility patters, thus risking a sub-optimal MDA outcome.

## Materials and methods

We developed an individual based, discrete time, spatially explicit, stochastic model, with mosquito population dynamics and human population movement. The flow diagram in *Figure 1A* describes the natural history of malaria infection in the human population. Details of how the dynamics of malaria transmission, human mobility and interventions are simulated are provided in the simulation protocol section below. All model parameters are presented in *Table 2* along with their respective references when applicable. The simulated synthetic population mimics the demographics of a set of 1000 villages in SE Asia, with the distribution of villages over space, village sizes and age distribution of people likely not applicable to African settings. They should be generic enough to give a fair representation of rural settings in SE Asia. The parameter exploration presented here provides the

**Table 2.** List of parameters used in the model.

| # | Name | Description | Value | Reference |
|---|------|-------------|-------|-----------|
| 1 | V | Set of villages | $|V| = 1000$ | |
| 2 | N | Set of individuals across all villages | $|N| = 314{,}795$ | - |
| 3 | ml | Proportion of males in the population | 0.48 | (**National Institute of Statistics, 2008**) |
| 4 | maxage | Maximum age | 80 years | - |
| 5 | beta | Biting rate | variable[*] | - |
| 6 | previ | Initial proportion of infectious people | variable[*] | - |
| 7 | resit | Initial proportion of artemisinin resistant infections | variable[*] | - |
| 8 | netuse | Proportion of individuals that own an insecticide treated bed net | 0.5 | - |
| 9 | itneffect | proportional decrease of individual susceptibility/ infectiousness related to ITN usage | 0.2 | - |
| 10 | ovstay | Mean number of nights spent somewhere when undertaking short-term population movement | 3 | - |
| 11 | crit | Critical distance below which overnight stays somewhere other than your home are made very unlikely | 4 km | - |
| 12 | timecomp | Mean time to complete ACT routine treatment | 4 days | Best guess |
| 13 | fullcourse | Proportion that receives treatment full course | 0.8 | (**Yeung et al., 2008**) |
| 14 | covab | Proportion of symptomatic cases that receive antimalarials | 0.6 | (**Yeung et al., 2008**) |
| 15 | nomp | Relative probability of receiving treatment in a non-malaria post village | 0.1 | Best guess |
| 16 | asymtreat | Relative probability of receiving treatment without clinical symptom | $10^{-4}$ | - |
| 17 | tauab | Daily probability of receiving ACT in a village under MDA | 1/1.5 | - |
| 18 | gamma | Mean liver stage duration | 5 days | (**Collins and Jeffery, 1999**; **Eyles and Young, 1951**) |
| 19 | sigma | Mean time to infectiousness after liver emergence | 15 days | (**Jeffery and Eyles, 1955**) |
| 20 | mellow | Mean duration of symptoms | 3 days | (**Church et al., 1997**) |
| 21 | xa0 | Daily probability of going below the minimum effective artemisinin concentration | 1/7 | (**Karbwang et al., 1998**) |
| 22 | xai | Daily probability losing the DHA effect as part of ACT | 1/3 | (**Rijken et al., 2011**; **Tarning et al., 2008**) |
| 23 | xab | Daily probability of going below the minimum effective piperaquine concentration | 1/30 | (**Rijken et al., 2011**; **Tarning et al., 2008**) |
| 24 | xpr | Daily probability of going below the minimum effective primaquine concentration | 1/2 | (**Burgess and Bray, 1961**) |
| 25 | delta | Mean duration of a malaria untreated infection | 160 days | (**Eyles and Young, 1951**; **Babiker et al., 1998**; **Franks et al., 2001**) |
| 26 | imm_min | Minimum clinical immunity period | 40 days | Best guess |
| 27 | alpha | Average permanence in each immunity level | 60 days | - |
| 28 | phic | Relative infectiousness of symptomatic infections compared to sub-patent ones | 1 | - |
| 29 | mdi | Mosquito daily probability of dying while infectious | 1/7 | (**Dawes et al., 2009**) |
| 30 | mdn | Mosquito daily probability of dying while infected but not yet infectious | 1/20 | (**Dawes et al., 2009**) |
| 31 | mgamma | Mean extrinsic incubation period | 14 days | (**Smith et al., 2014**) |
| 32 | amp | Amplitude of mosquito density seasonal variation | 0.6 | Best guess |

*Table 2 continued on next page*

Table 2 continued

| # | Name | Description | Value | Reference |
|---|------|-------------|-------|-----------|
| 33 | *process* | Days needed to administer a full ACT course in one village | 4 days | Optimistic guess |
| 34 | *rounds* | Number of drug rounds in an MDA campaign | 3 | Standard practice |
| 35 | *btrounds* | Number of days between drug rounds in an MDA campaign | 32 | Standard practice |
| 36 | *vcefficacy* | Vector control efficacy | variable[*] | - |
| 37 | $c_{b \cdot r0 \cdot a}$ | Daily probability of clearing blood stage drug sensitive parasites with circulating dha | 1/5 | (**Adjuik et al., 2004**; **Pukrittayakamee et al., 2004**) |
| 38 | $c_{b \cdot ra \cdot a}$ | Daily probability of clearing blood stage artemisinin resistant parasites with dha | $0.27 * c_{b \cdot r0 \cdot a}$ (0.05) | (**Dondorp et al., 2009**) |
| 39 | $c_{i \cdot r0 \cdot a}$ | Daily probability of clearing infectious stage drug sensitive parasites with circulating dha | 1/3 | (**Adjuik et al., 2004**; **Pukrittayakamee et al., 2004**) |
| 40 | $c_{i \cdot ra \cdot a}$ | Daily probability of clearing infectious stage artemisinin resistant parasites with dha | $0.27 * c_{i \cdot r0 \cdot a}$ (0.09) | (**Dondorp et al., 2009**) |
| 41 | $c_{b \cdot r0 \cdot ab}$ | Daily probability of clearing blood stage drug sensitive parasites with circulating dha- piperaquine | 1/3 | (**Adjuik et al., 2004**; **Pukrittayakamee et al., 2004**) |
| 42 | $c_{b \cdot ra \cdot ab}$ | Daily probability of clearing blood stage artemisinin resistant parasites with dha- piperaquine | $0.27 * c_{b \cdot r0 \cdot ab} + (1.0–0.27) * c_{b \cdot r0 \cdot b}$ (0.33) | (**Dondorp et al., 2009**) |
| 43 | $c_{i \cdot r0 \cdot ab}$ | Daily probability of clearing infectious stage drug sensitive parasites with circulating dha- piperaquine | 1/3 | (**Bustos et al., 2013**) |
| 44 | $c_{i \cdot ra \cdot ab}$ | Daily probability of clearing infectious stage artemisinin resistant parasites with dha- piperaquine | $0.27 * c_{i \cdot r0 \cdot ab} + (1.0–0.27) * c_{i \cdot r0 \cdot b}$ (0.126) | - |
| 45 | $c_{b \cdot r0 \cdot b}$ | Daily probability of clearing blood stage drug sensitive parasites with circulating piperaquine | 1/3 | (**Chen et al., 1982**) |
| 46 | $c_{b \cdot ra \cdot b}$ | Daily probability of clearing blood stage artemisinin resistant parasites with piperaquine | 1/3 | (**Chen et al., 1982**) |
| 47 | $c_{i \cdot r0 \cdot b}$ | Daily probability of clearing infectious stage drug sensitive parasites with circulating piperaquine | 1/20 | (**Myint et al., 2007**) |
| 48 | $c_{i \cdot ra \cdot b}$ | Daily probability of clearing infectious stage artemisinin resistant parasites with piperaquine | 1/20 | (**Myint et al., 2007**) |
| 49 | $c_{i \cdot r0 \cdot p}$ | Daily probability of clearing infectious stage drug sensitive parasites with primaquine | 1/1.5 | (**Burgess and Bray, 1961**; **Smithuis et al., 2010**) |
| 50 | $c_{i \cdot ra \cdot p}$ | Daily probability of clearing infectious stage artemisinin resistant parasites with primaquine | 1/1.5 | - |
| 51 | $k$ | Steepness of susceptibility increase with age | 0.14 | (**Aguas et al., 2008**) |
| 52 | $r$ | Amplitude of susceptibility increase with age | 0.99 | (**Aguas et al., 2008**) |

[*]the values are varied in different simulation settings. Their values are given in the description of each set of experiments and the set of possible values is given in **Table 1**.

limits of plausibility in terms of how people are expected to move, the levels of drug resistance and malaria prevalence, and how spatially heterogeneous transmission is.

Spatial demographics for both mosquitoes and humans are implemented at the village level, that is, humans can only move between villages and transmission within each village follows a pseudo-homogenous process where each mosquito is equally likely to bite a given individual. We assume villages are transmission units that encapsulate the village itself and the surrounding farms/ forest areas. Whilst clearly a simplification of reality, we know that mosquitoes can easily cover the distance between village and proximal farms over the course of a single day, and most SE-Asian vector species engage in late afternoon biting (some even having a near flat biting rate throughout the day). Given that people tend to move freely within their village during those hours we can

reasonably assume that all humans in one village can potentially be bitten by any one mosquito in that village. Having villages as both the transmission and intervention units is obviously computationally convenient, since all model processes related to transmission and intervention can then be evaluated at the same scale.

Malaria transmission in the Greater Mekong Sub-region (GMS) has been reported to be spatially heterogeneous (**Gryseels et al., 2015**; **Cui et al., 2012**; **Erhart et al., 2005**). However, there is very little evidence as to the relative abundance of the main vector species and no quantification of densities exists at a large enough scale. Given the lack of data to inform the discrepancies in mosquito densities across different villages, we chose to explore three mosquito density distributions (*Mn*, *Ml*, and *Mp*). Two of those distributions represent extreme scenarios: one in which all villages have approximately the same biting rate (Gaussian distribution); another where the vast majority has very low biting rates with a few hotspots (Pareto). The third distribution illustrates a scenario possibly more applicable to most areas in which some villages have a quite high transmission intensity, but where most have low mosquito abundance. These distributions were chosen arbitrarily and are parameterised as follows:

$$Gaussian:$$
$$Mn \sim N(0.0172, 0.0075)$$

$$Lognormal:$$
$$Ml \sim Logn(\mu, \sigma)$$
$$\mu = Log\left(\frac{0.0172^2}{\sqrt{(0.0172^2 + 0.0001)}}\right)$$
$$\sigma = \sqrt{Log\left(\frac{0.0001}{0.0172^2} + 1\right)}$$

$$Pareto$$
$$Mp \sim Pareto(0.37, 0.0075, 0.001)$$

We take these distributions of mosquito density across villages as reference ($M(t=0)$) and impose some seasonal variation to reflect the observed malaria incidence seasonal patterns. Mosquito density at time $t$ in village $j$, for all mosquito density distributions is then given by:

$$M_i(t) = M_i(0) + amp \times M_i(0) \times \cos\left(2\pi\left(\frac{t-90}{365}\right)\right),$$

where *amp* reflects the amplitude of the seasonal fluctuation.

Transmission in a given simulation is characterised by a single extra parameter, the mosquito biting rate, which determines the vectorial capacity and is adjusted to reach a specific baseline malaria prevalence in the human population. The mosquito biting rate calibration was performed for each combination of transmission heterogeneity distribution, population mobility intensity and malaria mean prevalence. Thus, for each population mobility and mosquito density distribution, hundreds of model runs were performed until the desired mean malaria prevalence was reached with a specific mosquito biting rate within the first 5 years of simulation.

Population movement patterns and their importance for infectious disease transmission and emergence has recently garnered a lot of increased scientific interest, with new tools and analysis frameworks being developed for mobility inference (**Tatem et al., 2014**; **Tatem et al., 2009**). Whilst census data and mobile phone data can help in proposing a connectivity network for a given region, the lack of general precision in questionnaires and the relative difficulty in capturing a lot of outside home overnight stays in mobile phone records, begs for a new source of data to resolve transmission relevant mobility patterns. Spatially explicit malaria models have in so far used gravity models to describe population movement, which is supported by some data (notably, daytime travel data). Whilst we agree that a gravity model can be the most appropriate to describe some seasonal and long-term migration patterns, we argue that overnight stays a very short distance from your home are generally unlikely. It is more likely for someone to return home for the night if they are within a certain critical distance threshold (*crit*) in km, instead of staying overnight somewhere else. We thus consider that daily population flow (*FL_short*) between villages $i$ and $j$ is best characterised by a modified gravity model given by:

$$FL\_short_{ij} = FL_{ij} / \left(1 + \frac{1}{1 + e^{10 \times (dist(i,j) - crit)}}\right)$$

where $FL_{ij}$ refers to the daily population flow of a standard gravity model

$$FL_{ij} = \left(\frac{pop_i \times pop_j}{\sqrt{dist(i,j)}}\right)$$

where $pop$ refers to village population size, and $dist$ to the Euclidean distance (in km) between villages $i$ and $j$.

The frequency of general population's short-term movements (very small number of nights spent somewhere other than their own village at a time) is given by the overall population mobility parameter – $mobility$ – which is explored at length in the main text and assumes 2 extreme values (5 and 250 nights spent somewhere other than the home village, per year). The population is further partitioned into temporary (seasonal) or long-term migrant groups. Seasonal migration can only occur during a 3 month period. This roughly corresponds to the duration of crop seasons, at which time people typically go back to their village of origin to help their families harvest crops or for other economic/personal reasons. After 3 months they return to larger villages or surrounding cities following $FL_{ij}$. Long-term migrants only move between a priori defined (at random) economic hubs, comprising 2% of the target villages, spending an average of 6 months in each before moving to the next. For simplicity, we only explore the effects of short-term movements throughout this paper, excluding seasonal and long-term migration events from the simulations presented here. We should note that the sensitivity of the model's predictions to seasonal and long-term migration is much lower than to the general population short-term movements explored to great lengths throughout this paper. We simulate malaria elimination strategies composed of MDA, a village malaria worker (VMW) network for improved case management, and an annual bed net distribution program. Villages are given an MDA of one full course (3 monthly rounds) of artemisinin combination therapy (ACT) plus one dose of primaquine, irrespective of their illness or infection status. Logistically, intervention teams sweep through all villages and give out ACTs without any prior screening, staying for a given number of days and then moving on to the next village. The details of how implementation logistics were incorporated into the simulation protocol can be found in the Simulation Protocol below.

When exploring the factors driving MDA outcome prediction, we explored all possible combinations of parameter values presented in *Table 1*, comprising 3888 sets of parameters. The model was run 100 times for each parameter set.

We also explored the layering of further intervention efforts on top of a global MDA initiative (with 3 ACT rounds), such as the implementation of indoor residual spraying (IRS) or larvicidal deployment, for example. Vector control effect size can be modelled as a reduction in vectorial capacity or EIR, as a direct consequence of a decrease in life expectancy, increase in sporogony cycle length and/or decreased biting rate on humans. Depending on what vector control measure one considers and how the mosquito life cycle is modelled, there might be interest in detailing the impact of a vector control intervention on a particular aspect of the mosquito life cycle. That is beyond the scope of what is intended in this paper, and we present vector control effect size as a measure of how much vectorial capacity is decreased when vector control interventions are in place. Thus, a transmission reduction efficacy of 0.10 means that the implemented vector control strategy reduces the number of infectious bites per person per year by 10%. Whilst MDA is programmatically well defined, with a specific number of ACT rounds being deployed, it is less clear how vector control strategies are sustained over time. Thus, we ran simulations for a range of vector control efficacy/duration of intervention pairs and evaluated the proportion of simulations in which elimination is reached.

## Simulation protocol

In this section, we give detailed description of the agent-based malaria simulation model results from which were discussed in the main text. We start by defining the interacting agents and their properties in the next section. Model processes and functions executed during the simulation are then documented in another two sections according to their positions in the simulation sequence.

## Agents

The simulation model presented here is a multi-level agent-based model containing three interconnected groups of agents: Villages, Humans and Mosquitoes. Each agent has group-specific properties:

## Villages

- Village ID
- Population size
- Location (Longitude, Latitude)
- Mosquito density
- Malaria post status/Date of establishment
- Current treatment strategy (whether the village is undergoing MDA)
- Number of administered ACT rounds

## Humans

- Human ID
- Home Village ID
- Current Village ID
- Age
- Gender
- Susceptibility/infectiousness
- ITN usage (affects the susceptibility/infectiousness property above)
- List of infections (In each human we keep a list of all infections emerging from the liver. We track each infection's parasite drug resistance status and maturity over time)
- Transmission status
- Clinical status
- Immunity status (Immunity level and cumulative number of lifetime infections determined the probability of developing clinical symptoms upon infection)
- Treatment status
- Active circulating drugs (which drugs are circulating at effective concentrations in the person's blood)

## Mosquitoes

- Transmission status
- Infection carried (This property informs on the drug resistance of the parasites in the mosquito's salivary glands)
- Current Village ID

## Model set-up

To explore the effect of malaria interventions using dynamic transmission models, one usually assures that the model is run until an equilibrium is reached (thus establishing a control scenario), and then implements whatever the intervention of interest is. The intervention's outcome or effectiveness can then be derived from a direct comparison between the integral of the control scenario and that of the intervention scenario over the same time period. Throughout the manuscript we present simulations for settings of a specific malaria prevalence. That value is the mean malaria prevalence over the last year of a 200 year run of a control scenario. To calibrate model runs to a specific prevalence we vary a free parameter *beta* [5], the mosquito biting rate, whilst keeping all other parameters fixed.

From the end of each successful calibration run to a given prevalence, we extract individual level information to inform the age, immunity level, number of cumulative infections, and infection status of each individual. This information forms a human input file used for model initialisation in the runs where intervention is simulated. The calibrated vectorial capacity distributions used in the calibrated runs are added to a village input file containing village location and population size and is also used for model initialisation.

More specifically, the following processes are involved in setting up the human population and village properties at the start of the 200 year prevalence calibration runs:

## Human properties

### Gender and age

Age and gender information of human agents are generated according to parameterised distributions. The probability of a human agent being male is *ml* [3]. The age of a human is sampled from a discrete distribution specified by a vector of size *maxage* [4]. This vector is given by a csv file with *maxage* [4] integers, containing a discrete age profile taken from Cambodian census data.

### Prevalence

The probability of a human agent being infectious is *previ* [6], and the probability of that infection being resistant to artemisinin is *resit* [7].

### ITN usage

Each human agent is assigned with an ITN with a probability of *netuse* [8]. Ownership of an ITN reduces the human agent's susceptibility/infectivity by *itneffect* [9]. Note that ITN distribution is not explored further in the model runs contained here. If that were a consideration, then the control scenarios would have both *netuse* and *itneffect* at their minimum acceptable values.

### Immunity

Each human agent is assigned two properties in relation to immunity, namely Cumulative Number of Exposures and Immunity Level. Both properties are set to 0 for newborns. The likelihood of clinical symptoms brought on by a single infection is given by

$$clinical\_prob_n = e^{(-0.15 \times (moi_n - 1))} \times \frac{0.1 \times e^{-(cml_n - 2) \times 0.1} + e^{-0.9 \times cml_n}}{lvl_n^{0.5}}$$

where *moi*, *cml* and *lvl* denote the multiplicity of infection, cumulative exposure to malaria and immunity level properties of the human agent respectively. Although we only increase immunity level if individuals resolve their infection (presumably due to increased antibodies killing activity), the cumulative exposure is updated with each infectious bite received. Thus, individuals can accrue some immunity with superinfections.

## Village properties

### Mobility network

Geo-spatial human mobility amongst the population is a key element simulated in our model. Using the location and population information provided for each village, a complete graph (*FL*) is constructed linking all villages during initialisation. Let *M* denote the set of villages in the simulation, *FL* is constructed using a generic gravity model, with each edge denoting the flow of human movements between village $i, j \in V$ as

$$FL_{ij} = \left( \frac{pop_i \times pop_j}{\sqrt{dist(i,j)}} \right)$$

where $pop_i$ denotes the population of village $i$, and $dist(i,j)$ denotes the earth-surface distance between $i$ and $j$.

### Mosquito density

Each village has a property describing the number of mosquitoes per person. At model setup, each village is assigned a mosquito density by randomly sampling from the distribution describing the spatial heterogeneity in risk used for a particular model run. Given the lack of data to inform these distributions, we chose to explore three different ones (*Mn*, *Ml*, and *Mp*). Two of those distributions represent extreme scenarios: one in which all villages have approximately the same biting rate (Gaussian); another where the vast majority have very low biting rates with only a few hotspots

(Pareto). The third distribution (Lognormal) illustrates an intermediate scenario in which some villages have a quite high transmission intensity, but where most have low mosquito abundance. These distributions were chosen arbitrarily and are parameterised as follows:

$$Gaussian : Mn \sim N(0.0172, 0.0075)$$

$$Lognormal : Ml \sim Logn(\mu, \sigma)$$

$$\mu = \text{Log}\left(\frac{0.0172^2}{\sqrt{(0.0172^2 + 0.0001)}}\right)$$

$$\sigma = \sqrt{\text{Log}\left(\frac{0.0001}{0.0172^2} + 1\right)}$$

$$Pareto : Mp \sim Pareto(0.37, 0.0075, 0.001)$$

We take these distributions of mosquito density across villages as reference ($M(t = 0)$) and impose some seasonal variation to reflect the observed malaria incidence seasonal patterns (described below).

For simplicity, we chose not to create infectious mosquitoes during model set-up. This is essentially due to the extremely fast timeframes of life events in mosquitoes compared to humans. The mosquito infection prevalence reaches equilibrium in a matter of days and thus its initialisation to non-zero values bears a negligible benefit. We use a free parameter *beta* [5], the mosquito biting rate, to calibrate each simulation to the desired malaria prevalence. Biting rates are calibrated for each combination of mobility, transmission distribution, and prevalence to ensure the system is at equilibrium.

## Model initialisation

The developed malaria micro-simulation platform takes inputs from two CSV-formatted input files and a JSON-formatted configuration file. The two input files provide the model with a list of villages and a list of humans respectively. Each row in these files describes the properties (as listed in the previous section) of either a village or a human agent. Model-wide and process-specific, as opposed to agent-specific, parameters are given by the configuration file. Most parameters specified in *Table 2* of the main text are associated with processes and functions (rather than individual agents), and therefore are given by the configuration file. In this section, we use the numbers in square brackets to refer to the associated parameter number whose description and value can be found in *Table 2*.

The initialisation process starts by processing the configuration file where the location of the input files is stored. Then a list of village agents and a list of human agents are created according to information given in the input files. Human agents are randomly assigned a home village from a list of all possible villages. Once all agents have been created, the software initialises the parameterised functions of the model using the information given in the rest of the configuration file.

## Implementation of malaria relevant dynamics

Once initialisation finishes, the main body of the model simulation starts. We assume time zero to be the 1st of January 5 years prior to the first malaria post establishment/MDA initiation.

The simulation protocols detailed in this section illustrate the flow of events and processes taking place each day. Each process has a daily probability of occurrence. For each individual and for every simulated daily time step, a uniform random number is drawn between 0 and 1 for each possible and valid (according to the individual's status) transition process (e.g. whether the individual dies, is infected, is treated, etc) associated with that individual. The transition process occurs if and only if the number drawn is lower than its daily probability of occurrence. Note that not all events are valid to an individual given its status on a given day. For instance, a clinical resolution event is not valid to an individual until that individual develops clinical symptoms.

## Human population dynamics

### Birth and death

The age profile presented here are taken from Cambodia census data. When we initialize individuals in the model and assign them a random age given the age frequencies in the data (through representative sampling), and run that model for 200 years, whilst assuming that individuals have a life expectancy of 1/mu, we end up with a different age profile from the one we started with. That means an adjustment in life expectancy is needed to reflect differences in mortality rates across ages. To that end, we fit the data age profile to an 8[th] order polynomial function, which we normalised to obtain representation weights for each age, $w(a)$. We then calculate the death probability of individual $n$ of age $a$ as:

$$death\_prob_n(a) = \frac{1}{80-a} w(a)$$

Whilst this provided a large improvement is the long-term age profiles produced by the model, we still good not replicate the flattening around age 30. Research into the census data for LMICs revealed that there is a significant drop in life expectancy in teenager and young adults, presumably due to involvement in higher risk activities. We explored several age range mortality modifiers and found that the data is best fit when multiplying $w(a_{15-25})$ by 4.

When a death event happens, a new infant agent is generated as a replacement. The new agent is placed in the same village where the death took place to keep population size constant, and all its immunity and exposure related parameters reset.

### Mobility

Every human agent can display short-term mobility patterns, characterised by overnight stays in villages other than their home for a mean period of *ovstay* [10] days. For every agent who is currently located at their home village, the daily probability of such short-term movement is given by the factor *Mobility* in **Table 1**. Thus, the number of human agents embarking on short-term movement on a given day is

$$N_{move} \sim B(N_{home}, Mobility)$$

The destination of each agent's movement varies and is determined using the mobility network $FL$ constructed during initialisation. Let the flow of short-term movement between village $i, j \in V$ be

$$FL\_short_{ij} = FL_{ij} / \left( 1 + \frac{1}{1 + e^{10 \times (dist(i,j) - crit)}} \right)$$

where *crit* [11] denotes the critical distance below which overnight stays at a village other than home are made very unlikely. The probability of a human agent to move from *home* to village $j$ is

$$short\_move\_prob_{home,j} = FL\_short_{home,j} / \sum_{v \in V} FL\_short_{home,v}$$

### Clinical outcome

The likelihood of clinical symptoms brought on by a single infection is given by

$$clinical\_prob_n = e^{(-0.15 \times (moi_n - 1))} \times \frac{0.1 \times e^{-(cml_n - 2) \times 0.1} + e^{-0.9 \times cml_n}}{lvl_n^{0.5}}$$

where *moi*, *cml* and *lvl* denote the multiplicity of infection, cumulative exposure to malaria and immunity level properties of the human agent respectively.

### Treatment

The probability for a human agent to receive a full course of ACT treatment is dependent on symptomatology as well as the presence of a local malaria post. In a village with a malaria post, a human agent with clinical symptom would receive treatment with probability

$$treatment\_prob\_mp\_clinical = \frac{1}{timecomp} \times fullcourse \times covab$$

where *timecomp* [12], *fullcourse* [13] and *covab* [14] are described in *Table 2*. In a village with no malaria worker presence, this treatment probability is reduced by *nomp* [15]. Treatment rates of asymptomatic human agent is negligible as treatment is conditional on a positive RDT, which is very unlikely in sub-patent infections. The probability of a human agent with asymptomatic infection getting treatment is given by *asymtreat* [16]. The assumed figure of 60% treatment coverage may seem low but it was the coverage reported in a very comprehensive study designed specifically to evaluate access to treatment in remote areas covered by the Cambodian village malaria worker network and/ or malaria outreach teams (*Yeung et al., 2008*) Nevertheless, in Cambodia, the Thai-Myanmar border and other areas of Myanmar (*Landier et al., 2018*) there has been a substantial decrease in incidence over the past 5 years, mostly due to better clinical case management. The model presented here does predict a substantial effect size on prevalence when village malaria posts are open for this low of a coverage value (~20% decrease for starting prevalence of 5%).

In a village under MDA, the daily probability for a human agent to receive a round of ACT treatment is *tauab* [17].

## Intrinsic incubation
Parasites emerge from the liver at a rate of 1/*gamma* [18], thus the liver stage takes on average *gamma* [18] days to complete.

## Gametocytaemia
Parasites start reproducing sexually, and thus generating gametocytes with 1/*sigma* [19] daily probability.

## Clinical resolution
Malaria induced fevers gradually recede at a rate of 1/*mellow* [20], meaning that a person is feverish for *mellow* [25] days on average.

## PK/PD
We describe waning drug efficacy over time through explicit daily probabilities of drug effect loss. Upon receiving treatment (with DHA-pip), each human agent will gradually lose the effect of both DHA and Piperaquine, according to *xai* [22] and *xab* [23] respectively. The single remaining drug will be lost at rates *xa0* [21] and *xab* [23] for Artesunate and Piperaquine respectively. Single dose Primaquine is lost at rate *xpr* [24].

Parasite killing rates depend on the person's transmission status ($s \in \{\boldsymbol{blood}, \boldsymbol{infectious}\}$), with parasite clearance in not yet infectious people generally slower than that in individuals carrying gametocytes. Clearance of parasites with drug resistance phenotype $h$ by drug $d$ then follows

$$Clearance_{s \cdot h \cdot d} \sim B(N_{s \cdot h \cdot d}, c_{s \cdot h \cdot d})$$

where $c_{s \cdot h \cdot d}$ is an element of a 3-dimensional drug clearance rate matrix $C$ of size $|S| \times |H| \times |D|$. Values of the elements of $C$ are given by parameters [37-50] in Table 2. $B$ denotes a binomial distribution.

## Recovery
Each infection in a human agent's infection list has a daily probability of being naturally cleared given by 1/*delta* [25].

## Immunity
One level of clinical immunity is gained by a human agent every time his infection list is emptied. Immunity loss starts *imm_min* [26] days after one level of immunity is gained. Immunity is lost at a rate of 1/*alpha* [27]. Therefore, each human agent is clinically immune an average of *imm_min* [26] +

*alpha* [27] days. A loss in immunity prompts a reduction in immunity level and not the immune status per se.

## Susceptibility/Infectiousness

Susceptibility was implemented as being age dependent and be modulated by ITN usage. Each individual's baseline susceptibility increases during the first years of life, saturating at around age 10 (*Smith et al., 2004*). We define the susceptibility of individuals with age *a* to a mosquito bite as:

$$\delta(a) = 1 - r * exp(-k * a));$$

If someone sleeps under a bed net, their susceptibility to receive an infectious mosquito bite is reduced by *itneffect* [9]. Individuals sleeping under a bed net are also less infectious compared to people that don't.

Clinical status modulates infectiousness through *phic* [28], which determines the relative infectiousness of clinical malaria infections compared to sub-patent ones (here assumed to be 1). We recently published a paper stressing the need for further empirical studies to better characterise this critical but very controversial quantity (*Aguas et al., 2018*), where we demonstrate how the relative infectivity of chronic infections has severe consequences for malaria elimination prospects.

## Infection

Given the time dependent vectorial capacity of each village, we can extrapolate the number of mosquito bites landed on humans each day. We exclude all bites from non-infectious mosquitoes. Infectious bites are distributed across humans according to a Gaussian distribution of mean of 1 and a standard deviation of 0.5, reflecting how some individuals are more likely to be bitten than others. This is done through proportional sampling. For each infectious bite, the probability of causing a new infection in a human agent *n* is given by

$$Infection_n \sim B(1, susceptibility_n)$$

A resulting infection is then added to the human agent's infection list and inherits the drug resistance phenotype of the infecting mosquito. The number of that agent's cumulative number of exposures and multiplicity of infection is adjusted accordingly.

## Mosquito dynamics

Note that only infected and infectious mosquito agents exist in our model.

### Survival

We assume adult female mosquito's life expectancy to be 1/*mdi* [29] + 1/*mdn* [30] days on average. Meaning for each mosquito agent, its daily probability of dying while infectious is 1/*mdi* [29], and 1/*mdn* [30] while infected but not infectious.

### Extrinsic incubation

Parasite development in mosquitoes takes an average of *mgamma* [31] days. Meaning for a mosquito agent, it takes *mgamma* [31] days from gametocyte infection to having sporozoites in the salivary glands and thus becoming infectious to humans.

### Seasonality

Mosquito density at time *t* in village *i*, for all mosquito density distributions follows an annual seasonal cycle given by

$$M_i(t) = M_i(0) + amp \times M_i(0) \times \cos\left(2\pi\left(\frac{t-90}{365}\right)\right)$$

when only 1 seasonal peak is modelled in a year, and

$$M_i(t) = M_i(0) + amp \times M_i(0) \times \cos\left(4\pi\left(\frac{t-40}{365}\right)\right)$$

when 2 seasonal peaks are modelled in one year, with amp [32] denoting the amplitude of the seasonal variation of mosquito density.

## Infection

Given the time dependent vectorial capacity for each village, we can easily extrapolate the number of mosquito bites landed on humans each day. For all bites handed out by mosquitoes that land on infectious humans we generate a new infection in the corresponding mosquito with the same resistant phenotype as a randomly sampled infection in the human's infected list. There is no limit as to the number of infections a given mosquito can acquire during its lifetime. Their adult survivorship is quite limited though, with only 26% of female mosquitoes having more than one infection at the time of death. That number drops to ~ 14% for values of prevalence amenable to elimination. Note that on a daily basis, the set of humans a mosquito can bite is assessed, based on the human movement across villages. If a human spends a night in village A, they are included in the set of possible humans bitten by mosquitoes in that village, even though that person might live in village B.

## Interventions

### Malaria post

Malaria posts are established over time by placing a village malaria worker upon MDA initiation in each village. The presence of a malaria post improves the access to treatment as mentioned in the Human treatment section above.

### MDA

In order to coordinate MDA teams to visit all villages without overlapping and repetition, a complete graph $VG$ connecting all villages to each other is first constructed. The weight of the edge between village $i$ and $j$ is given by $dist(i,j)$. This complete graph $VG$ is then reduced to its minimum spanning tree form, denoted $MST(VG)$, which we use to represent the road network connecting all villages.

Given that the MDA campaign includes $T$ teams (Table 1), $T$ starting locations are randomly selected from the nodes/villages of $MST(VG)$. Then, $T$ breadth-first-search algorithms are started from each of the $T$ starting locations. These search algorithms run simultaneously in coordinated rounds. Each round, an algorithm proposes the next village to be visited. Villages are added to the path of the algorithm which reaches it first. When an algorithm reaches a village that has been added/visited to the path of another algorithm, it continues searching until it finds an un-visited village in the same round. Once all villages have been visited, all algorithms stop. Each algorithm's path is used as the sequence of villages to be visited by each of the $T$ MDA teams.

An MDA team stays in a village for *process* [33] days for each one of the *rounds* [34] number of ACT courses administered. There are *btrounds* [35] days between drug rounds.

### Vector Control and targeted Vector Control

Each village's mosquito density is reduced by 1-*vcefficacy* [36] during vector control. During a targeted vector control campaign, only a selected subset of villages' mosquito density property is reduced.

## Model Calibration

The simulated synthetic population mimics the demographics of a set of 1000 villages in SE Asia. Whilst not using data from real villages, some properties describing the demographic fabric of the synthetic population are taken from SE Asian settings. Notably, the distribution of village population sizes and the shape of Euclidean distance network used to determine human population mobility are informed by data from the Thai-Myanmar border, whilst the age profile is extracted from the Cambodian population census. The population demographics simulated here should be generic enough to give a fair representation of rural settings in SE Asia but are not necessarily applicable to African settings.

To ensure the simulation platform produces outcomes that represent reasonable falciparum malaria transmission dynamics, we performed several model calibrations using malaria metadata. By doing so, we safeguard the generalizability of the results presented in the paper as well as the applicability of the model to any specific setting moving forward. The mentioned meta datasets describe

fundamental relationships between malariometric indices measured in as many different endemic settings as possible.

A preliminary model calibration done independently of this work, allowed the estimation of an age-dependent force of infection function. Using data from 8 endemic countries in sub-Saharan Africa, we were able to estimate how age modulates the risk of infection during the first years of life, saturating at around age 10 (*Aguas et al., 2008*). The susceptibility of individuals with age *a* to a mosquito bite is then given by:

$$\delta(a) = 1 - r * exp(-k * a));$$

Where *k* [51] determines how steeply susceptibility increases with age, and *r* [52] controls the amplitude of that increase. Given that this relationship is primarily the result of an increasing body weight and surface with age (*Smith et al., 2004*), and secondarily with the potentially increased protection conferred to infants and small children, we believe it is transferable to any other setting.

An initial model calibration was carried out to better characterise how immunity is developed over age, which remains a contentious and unresolved issue in falciparum malaria. To do so we use a dataset of immunity markers collected in 4 Cambodian sites, that describes seroconversion rates over time and extrapolates age profiles of clinical immunity (*Cook et al., 2012*). It is important to define what measured immunity means in this context. The data presents percent positivity of each collected specimen (hence single individual) as that defined by a cut-off of the mean optical density of the seronegative population plus three standard deviations (*Corran et al., 2008*). More importantly, this refers to measured $MSP-1_{19}$ antibodies which have demonstrated a significant association with a decrease in clinical falciparum malaria incidence (*Fowkes et al., 2010*), and display a very strong linear correlation with EIR ($R^2$ = 0.78) (*Corran et al., 2007*). Critically, a 15% reduction in symptomatic *P. falciparum* per doubling of antibody levels was observed (*Fowkes et al., 2010*), thus painting a picture of piecemeal acquisition of clinical immunity over age, and explaining the quasi-linear relationship with EIR. We take the seroconversion rates reported in this paper as the probability of immune positivity and translate that into a probability function governing the likelihood that a newly infected person of a given age will eventually develop clinical symptoms and thus account for a new clinical malaria case. We then calibrate our model to the prevalence metrics reported for each setting and output the age, number of cumulative infections and immunity level of each person at the last timestep of the model. If we define the probability of clinical outcome as:

$$probclin = (imm\_a * exp(-0.1(cml - 1)) + exp(-imm\_b * (cml + 1)))/ sqrt(lvl);$$

we can then jointly estimate the set of parameters $\theta = \{imm\_a, imm\_b\}$, that minimise the difference between the mean probability of clinical symptoms derived from the mean number of cumulative infections (*cml*) and mean immunity level (*lvl*) for each age category in the model output for each prevalence, and the measured probability of immunity (Pi). We thus minimise the following objective function:

$$f(\theta) = \sum_i \sqrt{(probclin(i) - Pi(i))^2}$$

where *i* are the ages in the dataset.

The resulting estimates of *imm_a* (0.5631 [0.4201–0.7061]) and *imm_b* (0.9652 [0.4976–1.433]) determine the shape of the probability of developing clinical symptoms relative to previous exposure as is depicted in *Figure 1C* of the main text. The model adjustment to the data can be visualised in *Figure 1—figure supplement 2*.

The second calibration performed was to the relationship between entomological inoculation rate (EIR) and falciparum malaria prevalence. The data shows a non-linear relationship between EIR and *Pf* prevalence measured by microscopy (*Guerra et al., 2007*) in over 90 endemic countries, that suggests a marked heterogeneity in individual infection risk (*Smith et al., 2005*). The relationship produced by the model described here is compared with the data in *Figure 1—figure supplement 3*. For a direct comparison with the data, we had to convert the prevalence obtained from the model (Pm) to a metric akin to the one obtained when using microscopy for diagnosis (Pd). We assume that an model equivalent of Pd can be derived from: Pm = *s*Pd + *sp*(1 − Pd), thus correcting for microscopy's sensitivity (*s*) and specificity (*sp*). We assumed these to be 85% and 96%, respectively. Note

that the model was not fit to the data. Rather, 1000 simulations were run using different mosquito biting rates (and thus EIR) and susceptibility distributions but keeping all other parameters (as defined here) fixed, and their outputs plotted against the data. Here we present model calibrations using a Gaussian bite receptivity distribution with mean of 1 and a standard deviation of 0.5 (See *Infection* section for more details). Note that the Pareto distribution for vector density across villages (blue dots) replicates the observed prevalence variance for a fixed EIR value much better than the Gaussian equivalent (red dots).

We opted to plot a point per village for each simulation, to try to reproduce the variance observed in the real data, instead of plotting a single median prevalence value per EIR. This enables us to explore how the relationship between EIR and prevalence can be modulated by factors such as the existence of a village malaria worker, or proximity to a high/low incidence village.

The third calibration assessed the relationship between *Pf* prevalence and clinical case incidence. This calibration is done to the subset of SE-Asian datapoints contained in the dataset published in *Patil et al. (2009)* and assumes the same relationship between true and measured prevalence as stated above. The incidence reported here is interpreted as the number of people with a febrile illness testing positive for *P. falciparum* independently of aetiology. We thus include a term (*asymtreat* [16]) to describe the proportion of malaria asymptomatic infections in the model outcome that might test *Pf* positive and have a concurrent fever of another aetiology, to adjust the model to the data. Here we present a calibrated model output assuming *asymtreat* = $10^{-4}$ (*Figure 1—figure supplement 4*).

The fourth model calibration is done in the absence of data, since we don't have access to a sufficiently detailed dataset to inform the relationship between age and prevalence in SE-Asia. Instead, we present the patterns generated by our model and compare them with outputs from other models where a rigorous fitting procedure to such data was performed (*Griffin et al., 2014*). *Figure 1—figure supplement 5* summarises how clinical cases are distributed across 4 age ranges according to prevalence.

Lastly, to assess the realism of the MDA implementations simulated throughout, we compare the predicted model outcomes after MDA, with those obtained in the MDA trials performed in the Thai-Myanmar border. We extracted the data on baseline and post MDA prevalence from a supplementary figure in *Landier et al. (2018)*. The model comparators are medians and percentiles of the prevalence in specific villages across one hundred simulations. The model was set up such that MDA would be done focally, that is, only villages with a prevalence over a certain threshold (5%) were eligible to receive an MDA. As per the trial protocol, during the year prior to MDA programme rollout, all 1000 villages were surveyed to establish a baseline prevalence. These surveys consisted of sampling 50 individuals from each village and perform a uPCR to determine their infection status. Here we assume that uPCR can detect 85% of all infections. Once a list of villages eligible to receive MDA is compiled, the model proceeds to compile a schedule for each village to be visited by an intervention team. This process is repeated 100 times, and for each village only the runs in which that village was selected for MDA are used to calculate the summary statistics. A comparison of the model's output with the empirical trial is presented in *Figure 1—figure supplement 6*.

## Acknowledgements

The authors would like to thank the members of the Malaria Elimination Task Force team and executive committee for their advice on the model structure and insights on the challenges of incorporating a large-scale targeted mass drug administration into a *P. falciparum* malaria elimination strategy.

## Additional information

### Funding

| Funder | Grant reference number | Author |
| --- | --- | --- |
| Bill and Melinda Gates Foundation | OPP1110500 | Sompob Saralamba<br>Yoel Lubell<br>Lisa J White<br>Ricardo Aguas |

| Bill and Melinda Gates Foundation | OPP1193472 | Bo Gao Ricardo Aguas |
| Wellcome | | Yoel Lubell Lisa J White Arjen M Dondorp |

The funders had no role in study design, data collection and interpretation, or the decision to submit the work for publication.

### Author contributions
Bo Gao, Software, Formal analysis, Methodology; Sompob Saralamba, Software, Methodology; Yoel Lubell, Conceptualization, Writing - review and editing; Lisa J White, Arjen M Dondorp, Conceptualization, Supervision; Ricardo Aguas, Conceptualization, Software, Formal analysis, Methodology

### Author ORCIDs
Bo Gao  https://orcid.org/0000-0002-7405-7507
Arjen M Dondorp  http://orcid.org/0000-0001-5190-2395
Ricardo Aguas  https://orcid.org/0000-0002-6507-6597

### Decision letter and Author response
Decision letter https://doi.org/10.7554/eLife.51773.sa1
Author response https://doi.org/10.7554/eLife.51773.sa2

## Additional files
### Supplementary files
• Transparent reporting form

### Data availability
The study presented here is purely theoretical and no data has been used apart from previously (and publicly) available data.

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
