## [Decision Letter]

**Acceptance summary:**

This work presents a new agent-based, spatial model of malaria transmission to explore determinants of mass drug administration success in Southeast Asia. It has the potential to inform policymaking towards malaria elimination in specific areas.

**Decision letter after peer review:**

Thank you for submitting your article "Not all MDAs should be created equal-determinants of MDA impact and designing MDAs towards malaria elimination" for consideration by *eLife*. Your article has been reviewed by three peer reviewers, and the evaluation has been overseen by Eduardo Franco, acting as Reviewing Editor and Senior Editor. The reviewers have opted to remain anonymous.

As is customary in *eLife*, the reviewers have discussed the reviews with one another. What follows below is my edited compilation of the essential and ancillary points provided by reviewers in their critiques and in their interaction post-review. Some of the reviewers' comments may seem to be simple queries or challenges that do not prompt revisions to the text. Please keep in mind, however, that readers may have the same perspective as the reviewer. Therefore, it is essential that you attempt to amend or expand the text to clarify the narrative accordingly.

Summary:

This work presents a new agent-based, spatial model of malaria transmission to explore determinants of mass drug administration (MDA) success in a Southeast Asian setting. The authors especially emphasize the following aspects as previously underexplored or unexplored in other theoretical and model-based MDA studies: the role of gradual implementation of MDA due to realistic logistical constraints of finite MDA teams and how drug resistance affects optimal choice of implementation options. Other factors such as initial prevalence, interaction of MDA with other interventions such as case management and vector control, and timing relative to the transmission season are also considered here as they have been considered by a diverse set of other models. In all, this work has the potential to inform policymaking towards malaria elimination in specific areas.

Title:

Except for well-recognized acronyms and abbreviations, technical terms must be spelled out in manuscript titles. Every acronym must be spelled out at the first instance it is used. Once introduced, it may be used throughout. This rule applies to the Abstract and main text independently. Do not introduce an acronym or abbreviation if it will not be used again (Abstract and main text considered independently). An acronym must represent a minimum of three words.

Essential revisions:

1) Issues related to the model formulation:

The authors' conclusions on speed of implementation and role of resistance in determining optimal implementation strategy are rather subtle points. At the same time, I believe this is a previously unpublished model and thus its introduction in the literature should also contain a rigorous demonstration of its ability to recapture field data, particularly epidemiological data. While there are plots in the supplemental material that show model outputs around some basic epidemiological quantities (incidence by age and prevalence, for example), I did not see any comparison with field data, which is surprising since the authors are close collaborators with excellent field researchers specializing in malaria in Southeast Asia. Absent these basic comparisons with field and clinical data, it is difficult to assess whether the model's structure and parametrization adequately capture key phenomena to the point that claims around speed of malaria resurgence after MDA, and response of wild type and resistance infections to drug treatments (and thus selection pressure on these parasites) are to be believed.

In addition, there appears to have been a lost opportunity to further capture actual malaria transmission and historical MDA implementation in this set of villages. While the authors consider 3 possible distributions of transmission intensity across their village population, it would also be helpful to parameterize each village's transmission intensity based on field data collected in these villages, or at least to compare the actual distribution against the 3 modelled options (especially since outcomes were somewhat dependent on the nature of this distribution). Similarly, it is unclear to what extent the modelled MDA implementation (selection of random villages to begin, then moving MDA teams to adjacent untreated villages) resembles the actual implementation of the MDA in Karen State. As a naïve outsider I would guess that teams perhaps started in villages at one end of the area and worked their way together toward the other end (?), which seems logistically simpler for supply chain and transportation. Since this paper is about making models of interventions more operationally realistic, it would be great to include even more realism.

Questions on model structure and parameterization:

I'm curious about the decision not to model non-infected, non-infectious mosquitoes and how that could affect model outcomes, particularly around vector control effect size.

On immunity, my understanding is that individual acquire immunity to clinical symptoms but no other kind of immunity, such as to high-density asexual infections. Thus, the infectiousness of asymptomatic individuals is identical. Is this an accurate assumption?

Similarly, the parameterization of symptomatic and asymptomatic infectiousness as equal is curious and seems to contradict some field data. Can the authors point to evidence that this is a reasonable choice for their setting?

The authors mention a 5-year initialization phase of their model and that to begin with, there is no immunity in the population. Looking at Figure 1—figure supplement 2, middle panel, it doesn't look to me like 5 years of initialization is sufficient to reach an equilibrated population immunity structure.

Case management rate in villages with malaria posts (0.6) seems low. I believe in many SE Asian villages, a clear reduction in malaria incidence is observable after the introduction of malaria posts, suggesting they have a considerable effect on reducing transmission. Does this also happen on the same timescale in the model, with case management rate as low as 0.6?

2) Issues related to interpretation:

My main comment is that the results presented did not quite show what is stated in the Abstract: "We conclude that mass drug interventions can be an invaluable tool towards malaria elimination in the right context, specifically when paired with effective vector control" because:

a) It overstates the impact of MDA given that the probability of malaria elimination is extremely small unless done in a very low transmission setting

b) To justify this conclusion, the authors need to show a slightly different outcome – the 'increase in probability of elimination when MDA is carried out'. I.e. there is the issue that in many models the probability of stochastic elimination of the parasite without any interventions is significant in low transmission areas. So, the absolute probability of elimination in the presence of MDA is not that informative, it's better to know the increase in probability. The authors need to check the probability of elimination without MDA in their model and present those results in comparison to simulations in which MDA is included, which I believe is achievable within the timeframe of the journal's revision process.

Another key point is that although the authors nicely include variation in exposure to malaria across different villages, it's well quantified that this variation exists also within a village and has a strong effect on R0 and elimination probability – was this included in the model? (e.g. see Goncalves 2018 *eLife:* https://www.ncbi.nlm.nih.gov/pubmed/29357976 and Woolhouse 1997 PNAS: https://www.ncbi.nlm.nih.gov/pubmed/8990210). It would be relatively straightforward to put this into an individual based model, by setting the force of infection to have a distribution around the mean, instead of being the same for each individual.

As a general comment the labelling of figures needs tightening up – e.g. lines, abbreviations, units on the axes are often not clearly defined (see below for specific comments). Ideally these should be understandable without going back to the main text and Materials and methods. It would be a shame if the interesting messages of the analysis did not get through because of being hard to interpret. Also, the model parameters need more justification and documentation in places (immunity, age structure, relative infectiousness of asymptomatic and symptomatic infections). It would be hard to reproduce the model from the description, although the code is provided. Some though not all of these could impact the MDA results.

3) Section-specific concerns:

The Abstract and Introduction read as if the paper is extremely general. Most of the time malaria, not a specific species, is discussed. In the second paragraph of the Introduction, falciparum is mentioned and many of the parameters seem to relate to falciparum. Furthermore, in the fifth paragraph the authors state that the "modular simulation platform that is customizable to any malaria transmission setting" but then in the Materials and methods first paragraph "likely not applicable to Africa."

Inconsistent terminology makes the results unclear.

– MDA rounds refer to two things in the paper. Results first paragraph, it seems "MDA rounds" means MDA campaigns. This occurs again in the Discussion second paragraph. At other times in the same paragraph, what appears to be campaigns are referred to as sets, as in this case sets of rounds are discussed. Is MDA rounds here related to campaigns or the 3 standard rounds of implementation.

– Results paragraph one, how does Figure 2—figure supplement 5 relate to artemisinin spread. Further, what is "outcome" in this figure? It is not clear from the text or caption.

– Results paragraph two, how is "a more static population" defined? Mobility is not well defined in the main text. A brief description is finally given in the second paragraph of the Discussion. Furthermore, mobility is described both as a fraction and as a decimal, which makes it difficult to compare across figures. How does connectivity relate to mobility? How does "well-connected" relate to mobility? Discussion paragraph four, does "low connectivity" mean low mobility?

– Results paragraph two, what constitutes "little difference"? Many of the curves in Figure 3—figure supplement 1 seem to be separated.

– In the same paragraph what are the "transmission heterogeneity distributions"? later in the paragraph "the other two" are mentioned but not named. They are not described anywhere clearly. Figure 4 is referenced here as a comparison between distributions but only uses a single distribution.

– At the end of paragraph two, how is the "significant correlation" measured? There appear to be no statistical tests mentioned.

– Results paragraph three, how has vector control "greatly improved"? Is this measured? Visually they appear similar.

– Discussion paragraph two, how are the mentioned factors a distant second and third? Where is this quantified?

– How is a slower MDA implementation optimal? The solid line often appears to exceed the dashed line.

More information is needed in the Materials and methods.

– In paragraph seven mobility is briefly defined but referred to the main text, where there is not a good description. How are the nights chosen in the simulation? How does seasonal and long-term migration occur in the simulation? Does this effect the biting rate in places?

– In subsection “Simulation Protocol”, what other co-infecting parasites are considered?

– What drugs are used?

– Subsection “Model Initialisation”, do the data describe the initial properties of things like age, transmission status, etc.?

– Subsection “Model Initialisation” paragraph two, two different implementations of how human agents are assigned is given, which is it?

– Subsection “Human Properties”, are cumulative number of exposures and immunity level set to 0 for humans of all ages? This seems unrealistic.

– Subsection “Village Properties” is the calibration of biting rate stable when you introduce mobility and change malaria prevalence?

– Subsection “Implementation of malaria relevant dynamics”, the implementation is unclear. A random number is drawn to see if there is an event and then random number are drawn to see if there are each of several events. Does this mean that an individual could be infected and treated and die all in the same day? If so, why go through the earlier event draw?

– Subsection “Human Population Dynamics”, when death happens, are individuals replaced by an infant or by an individual of any age? If the latter, how are their initial properties chosen.

– Subsection “Human Population Dynamics”, *moi, cml*, and *lvl* are not in Table 2. Could a table be created for parameters that change over time with an individual?

– Can individuals be infectious immediately with program 1/*sigma*? This seems unrealistic. Why not use a fixed time delay?

– “Parasite killing rates depend on the person’s transmission status (𝑠), with parasite clearance”, s is not in Table 2. Similar comment to above.

– “Susceptibility/Infectiousness”, why include *phic* if the value is set to 1 (no increased/decreased relative infectiousness)?

– Subsection “Mosquito Dynamics – Survival”, it appears these values are inverted.

– "uncommon for any female to reach multiparity" Is this quantified?

– How does immunity work? It's not discussed in detail, although some figures pertain to it.

Figures/Tables are hard to interpret due to missing information.

– Figure 1: In A what are *σ, ε,* and *ω*? In C, what are the immunity levels? Is 1 higher or lower than 4?

– Figure 2: x-axis is not labeled.

– Figure 3: Lots of things are varied but none are mentioned in the caption.

– Figure 4: What is varied is not evident from the caption, nor is the distribution apparent.

– Figure 1—figure supplement 1: Why cut off the distribution prior to 80 if that is the max?

– Figure 1—figure supplement 2: What is "mean immunity level"? Why does it go up and down with age?

– Figure 2—figure supplement 1: The size of the dots makes it hard to tell how far the lines go.

– Figure 2—figure supplement 2: What is outcome here?

– Figure 2—figure supplement 3: What is outcome here?

– Figure 2—figure supplement 5: What is outcome here?

– Table 2: What does variable mean for parameters such as *beta, previ, resit, vcefficacy*? Can these be listed somewhere?

– Table 2: Parameter 37 – 50 are never mentioned in the manuscript.

[Editors' note: further revisions were suggested prior to acceptance, as described below.]

Thank you for submitting your revised article "Not all MDAs should be created equal-determinants of MDA impact and designing MDAs towards malaria elimination" for consideration by *eLife*. Your article has been re-reviewed by the original peer reviewers, and overseen by Eduardo Franco as the Reviewing Editor and Senior Editor.

We are happy to see the effort you made at amending the paper to accommodate the concerns and suggestions from the reviewers. Once again, we are unable to accept it in its present form for publication. However, we are willing to consider a new revised version if you can address the additional concerns and suggestions below.

Essential revisions:

It is difficult to discern from the manuscript what things are functionally in the model and what are used in the analyses presented. These seem to be justified in the response to the reviewer but not clearly in the manuscript.

Attempt to capture realistic operational conditions, given that the paper is about the impact of actual operational conditions on MDA outcome. The MDA distribution structure and its implication on the authors' results are still not clear. The authors are adamant that they are not attempting to model an actual MDA distribution, but it seems that randomly selecting starting villages with which to seed MDA teams is unlikely to be how MDA is actually implemented, and potentially there are interactions between the spatial pattern of MDA implementation and the role of migration in MDA outcome. The authors must either show that their method of random selection of starting villages is how someone has operationalized MDA, or that a more operationally realistic village visitation order does not impact their findings.

Infectivity: it is true that there is much that we do not understand quantitatively about infectivity/infectiousness. However, the simplifying assumption that all infected individuals, symptomatic and asymptomatic, have the same infectivity seems a little strong, especially since the authors note that changing *phic* does impact the effectiveness of MDA. Might it not also impact how the outcome of the MDA depends on the logistical, demographic, and transmission factors explored in this paper?

---

## [Author Response]

Title:Except for well-recognized acronyms and abbreviations, technical terms must be spelled out in manuscript titles. Every acronym must be spelled out at the first instance it is used. Once introduced, it may be used throughout. This rule applies to the Abstract and main text independently. Do not introduce an acronym or abbreviation if it will not be used again (Abstract and main text considered independently). An acronym must represent a minimum of three words.

We appreciate the general reader might not be familiar with the term MDA, and thus have changed the first instance of its usage to Mass Drug Administration. In the interest of space, we have kept the other mentions of Mass Drug Administration to its acronym form but are happy to spell those out as well if need be.

Essential revisions:1) Issues related to the model formulation:The authors' conclusions on speed of implementation and role of resistance in determining optimal implementation strategy are rather subtle points. At the same time, I believe this is a previously unpublished model and thus its introduction in the literature should also contain a rigorous demonstration of its ability to recapture field data, particularly epidemiological data. While there are plots in the supplemental material that show model outputs around some basic epidemiological quantities (incidence by age and prevalence, for example), I did not see any comparison with field data, which is surprising since the authors are close collaborators with excellent field researchers specializing in malaria in Southeast Asia. Absent these basic comparisons with field and clinical data, it is difficult to assess whether the model's structure and parametrization adequately capture key phenomena to the point that claims around speed of malaria resurgence after MDA, and response of wild type and resistance infections to drug treatments (and thus selection pressure on these parasites) are to be believed.

We appreciate this issue being flagged, and believe the substantive pertinent changes made to the manuscript should now clarify how the model is able to recreate malaria phenomenology and key interactions between malariometric indices. We have extended the Simulation Protocol by including a very extensive model calibration section. This section includes model benchmarking of:

– EIR vs. prevalence

– Prevalence vs. clinical incidence

– Clinical malaria age profiles for varying prevalence

– Age profiles of clinical immunity

– MDA trials

We had not previously included this level of detail as the work presented in this paper is of a more theoretical nature and in parallel work, we are actually fitting a version of this model to very detailed data from SE Asia. We do appreciate that for any confidence to be had in the model outputs and projection contained here, a better anchoring on empirical data is imperative, and thus hope the additional calibrations provided will ease those concerns.

In addition, there appears to have been a lost opportunity to further capture actual malaria transmission and historical MDA implementation in this set of villages. While the authors consider 3 possible distributions of transmission intensity across their village population, it would also be helpful to parameterize each village's transmission intensity based on field data collected in these villages, or at least to compare the actual distribution against the 3 modeled options (especially since outcomes were somewhat dependent on the nature of this distribution). Similarly, it is unclear to what extent the modeled MDA implementation (selection of random villages to begin, then moving MDA teams to adjacent untreated villages) resembles the actual implementation of the MDA in Karen State. As a naïve outsider I would guess that teams perhaps started in villages at one end of the area and worked their way together toward the other end (?), which seems logistically simpler for supply chain and transportation. Since this paper is about making models of interventions more operationally realistic, it would be great to include even more realism.

This seems like an unfair criticism, building on knowledge of collaborations we have ongoing with partners in SE-Asia, which are outside of the remit of this paper. At no point do we mention this model is a model of MDA implementation in Karen State. In fact, we purposefully kept the setting generic, even mentioning the age profile was adapted from the Cambodian census, and that the demographics modelled here are typical of a rural SE Asian setting, but not necessarily an African one. We do not use any data from Karen State to parameterise the version of the model presented here, apart from mean household size and relative Euclidean distance between villages. We are working on a separate paper where the model presented here is fitted to Karen State data and will re-create the exact timing of MDA in each of the villages. Here, we had to develop a sophisticated graph traversal algorithm to design a schedule for MDA in each village which minimised time travelled from a given starting village. We have made a few changes in the manuscript to hopefully made it clearer that the work enclosed here explores the relationships between logistical aspects of MDA delivery and population mobility in settings of varying transmission dynamics (characterised by prevalence, seasonality patterns and heterogeneity in risk across space).

Questions on model structure and parameterization:I’m curious about the decision not to model non-infected, non-infectious mosquitoes and how that could affect model outcomes, particularly around vector control effect size.

Vector control effect size can be modelled as a reduction in vectorial capacity or EIR, as a direct consequence of a decrease in life expectancy, increase in sporogony cycle length and/or decreased biting rate on humans. Depending on what vector control measure one considers and how mosquito life cycles are modelled, there might be interest in detailing the impact of a vector control intervention on a particular aspect of the mosquito life cycle. That is beyond the scope of what is intended in this paper, and we present vector control effect size as a measure of how much vectorial capacity is decrease when a vector control intervention is in place. Given how we implemented the human/mosquito interface it is then straightforward that inclusion of susceptible mosquitoes is not necessary. Given a vectorial capacity, the number of new mosquito bites landing on humans per time step can be calculated. These bites are then allocated to adult mosquitoes which will be tracked over time as individual agents. This bears tremendous computational efficiency. We are currently preparing a technical paper for submission focusing exclusively on how this methodology performs compared to a typical ODE model and a full IBM model for the mosquito population both including susceptible mosquitoes as well. As the reviewer might appreciate that is in itself a lengthy standalone paper.

On immunity, my understanding is that individual acquire immunity to clinical symptoms but no other kind of immunity, such as to high-density asexual infections. Thus, the infectiousness of asymptomatic individuals is identical. Is this an accurate assumption?

We need to make a distinction between infectivity and infectiousness. Infectivity is the probability that an infectious person transmits its infection to a mosquito upon a single bite. Infectiousness is the overall transmissibility of one malaria infection over the course of the time during which the infected person is infective to mosquitoes. We believe the reviewer is referring to infectivity in this context. Since we do not track human parasite densities, we must make some simplifying assumptions. We recently published a paper stressing the need for further empirical studies to better characterise this critical but very controversial quantity (Aguas et al., 2018). There is a pervasive conflation in the literature of how asexual parasitaemia levels correlate with infectivity to mosquitoes, since the parasite forms which are actually transmitted to the mosquito are much harder to measure and hardly ever modelled. In fact, parasite densities seem fluctuate considerably over time as a consequence of the interplay between antigenic variation and corresponding host immune responses. In one study, the proportion of total parasite density comprising gametocytes was shown to be markedly age dependent (Ouedraogo et al., Malar. J. 2010; 9:281), suggesting that the proportion of gametocyte positive individuals without detectable concomitant asexual parasitaemia increases with age. Another study shows that gametocyte density can be larger in asymptomatic individuals than in some symptomatic patients (Gouagna et al. Parasitology 2004; 128:235–243). This last study also reported increased infectivity of asymptomatic gametocyte carriers not correlated with gametocyte densities, suggesting other factors such as gametocyte maturity and/or human blood factors might influence their infectivity. This might help explain why high gametocytes densities do not necessarily result in mosquito infections (Bousema et al. PLoS One 2012; 7, Schneider et al. Am. J. Trop. Med. Hyg. 2007; 76:470–474). The paper concludes that: “The relative infectivity of chronic infections has severe consequences for malaria elimination prospects. Reliable estimates to inform that parameter will require longitudinal follow-up studies of sufficient duration in both clinical and chronic infections, with serial mosquito-feeding assays. Although we appreciate the time-consuming efforts involved and the significant logistical burden (specifically on the insectary) required to perform such a study, we consider those concerns to be greatly outweighed by the dramatic expected impact on malaria elimination policy.”

To avoid getting into this controversy here, we have kept *phi_a_*=1. We have included a new reference to this paper for the reader’s benefit.

Similarly, the parameterization of symptomatic and asymptomatic infectiousness as equal is curious and seems to contradict some field data. Can the authors point to evidence that this is a reasonable choice for their setting?

Much of the justification for this is contained in the previous reply. However, we should add that assuming *phi_a_*=1 is not a conservative assumption, but rather a functional one. As explored in (Aguas et al., 2018,), higher values of *phi_a_* make the dynamics more amenable to control through mass scale drug-centric approaches, whereas for lower values of *phi_a_*, maximising clinical case management has a disproportionate impact whilst MDA style approaches are less effective. As this is already published, we opted to increase computational efficiency. Keeping in mind that our objective is not to accurately quantify the chances for malaria elimination in a single setting, but rather explore how the predicted outcome of MDA campaigns differs depending on several logistical, demographic and transmission factors; we chose the value of *phi_a_*that would decrease the amount of simulations needed to run for such comparisons to become possible. We have now included some more text in the Simulation Protocol to justify this choice.

The authors mention a 5-year initialization phase of their model and that to begin with, there is no immunity in the population. Looking at Figure 1—figure supplement 2, middle panel, it doesn’t look to me like 5 years of initialization is sufficient to reach an equilibrated population immunity structure.

We apologise for inadvertently raising confusion. We have included an extensive section on model set-up in the Simulation Protocol which provides a clearer idea of how equilibrium conditions are assured for each model simulation presented in the manuscript.

Case management rate in villages with malaria posts (0.6) seems low. I believe in many SE Asian villages, a clear reduction in malaria incidence is observable after the introduction of malaria posts, suggesting they have a considerable effect on reducing transmission. Does this also happen on the same timescale in the model, with case management rate as low as 0.6?

We agree that a coverage of 60% is far from optimal, however it was the reported coverage in a very comprehensive study designed specifically to evaluate access to treatment in remote areas covered by the Cambodian village malaria worker network and/or malaria outreach teams (Yeung et al., 2008). Nevertheless, in Cambodia, the Thai-Myanmar border and other areas of Myanmar there has been a substantial decrease in incidence over the past 5 years, mostly due to better clinical case management. The model presented here does predict a substantial effect size on prevalence when village malaria posts are open for this low of a coverage value (~20% decrease for starting prevalence of 5%), but more work is needed to disentangle the true effect size of the malaria post network from the data itself.

The settings adopting this strategy had concurrent interventions, do not have rigorous estimates for village population sizes, and cannot account for treatment seeking behaviours in the private sector (even in remote areas of the GMS one can usually find a private chemist). As mentioned before, we are working hard to fit a version of the model presented here to incidence data in the Thai-Myanmar border which might lead to a better understanding of these processes through estimation of treatment seeking related parameters.

2) Issues related to interpretation:My main comment is that the results presented did not quite show what is stated in the Abstract: “We conclude that mass drug interventions can be an invaluable tool towards malaria elimination in the right context, specifically when paired with effective vector control” because:a) It overstates the impact of MDA given that the probability of malaria elimination is extremely small unless done in a very low transmission setting.

We were cautious to include the safeguard “in the right context” to assuage the previous qualification. The answer to the next point might satisfy the reviewer that MDA is a requirement for elimination to be reached in such a short time scale even in very low prevalence settings. Nonetheless, we have made it much clearer in the manuscript that elimination is predicted to be very unlikely in all but very low transmission settings.

b) To justify this conclusion, the authors need to show a slightly different outcome – the ‘increase in probability of elimination when MDA is carried out’. I.e. there is the issue that in many models the probability of stochastic elimination of the parasite without any interventions is significant in low transmission areas. So the absolute probability of elimination in the presence of MDA is not that informative, it’s better to know the increase in probability. The authors need to check the probability of elimination without MDA in their model and present those results in comparison to simulations in which MDA is included, which I believe is achievable within the timeframe of the journal’s revision process.

We appreciate the suggestion and have performed the suggested simulations. The results are presented in Figure 4—figure supplement 3. It is strikingly clear that although a significant reduction in prevalence can be achieve with the 5-year time frame, elimination without MDA is incredibly unlikely. In fact, none of the 100 runs for each parameter set resulted in elimination. Thus, the previously presented value is the increase in probability, since the probability is zero for vector control only. These effects had been previously explored by us in (Aguas et al., 2018) and can be modulated by *phi_a_*and changes in baseline treatment rates.

Another key point is that although the authors nicely include variation in exposure to malaria across different villages, it’s well quantified that this variation exists also within a village and has a strong effect on R0 and elimination probability – was this included in the model? (e.g. see Goncalves 2018 eLife https://www.ncbi.nlm.nih.gov/pubmed/29357976 and Woolhouse 1997 PNAS https://www.ncbi.nlm.nih.gov/pubmed/8990210). It would be relatively straightforward to put this into an individual based model, by setting the force of infection to have a distribution around the mean, instead of being the same for each individual.

This is one of the aspects that was not addressed in the original submission but gained prominence as we detailed the model calibration process. We have indeed incorporated different distributions to describe differential biting “attractiveness”, i.e., describing how mosquito bites are distributed across the human population. Given the model calibration to the EIR vs. prevalence metadata, we kept with a Gaussian distribution, with mean 1 and standard deviation of 0.5.

As a general comment the labelling of figures needs tightening up – e.g. lines, abbreviations, units on the axes are often not clearly defined (see below for specific comments). Ideally these should be understandable without going back to the main text and Materials and methods. It would be a shame if the interesting messages of the analysis did not get through because of being hard to interpret. Also, the model parameters need more justification and documentation in places (immunity, age structure, relative infectiousness of asymptomatic and symptomatic infections). It would be hard to reproduce the model from the description, although the code is provided. Some though not all of these could impact the MDA results.

See detailed rebuttal to specific comments below.

3) Section-specific concerns:The Abstract and Introduction read as if the paper is extremely general. Most of the time malaria, not a specific species, is discussed. In the second paragraph of the Introduction, falciparum is mentioned and many of the parameters seem to relate to falciparum. Furthermore, in the fifth paragraph the authors state that the "modular simulation platform that is customizable to any malaria transmission setting" but then in the Materials and methods first paragraph "likely not applicable to Africa."

We acknowledge that the species we are referring to here should have been more at the forefront. We have hopefully rectified that in the revised manuscript. As for the second comment, we don’t see where a conflict exists and what is the source of confusion. The simulation platform is modular (functions can be removed and added without loss of function) and can be customised to any transmission setting. Because in this case we chose to customise it to SE-Asian settings (as further documented in the Materials and methods section) we cautioned against immediate applicability to African settings. By no means does that suppose we cannot customise the model to African settings and re-run the simulations presented here, but it is not in the scope of the current manuscript.

Inconsistent terminology makes the results unclear.– MDA rounds refer to two things in the paper. Results first paragraph, it seems "MDA rounds" means MDA campaigns. This occurs again in the Discussion second paragraph. At other times in the same paragraph, what appears to be campaigns are referred to as sets, as in this case sets of rounds are discussed. Is MDA rounds here related to campaigns or the 3 standard rounds of implementation.

We’ve corrected the main text at these given locations by replacing “rounds” and “set” with “campaign(s)” where referring to 3 rounds of MDA.

– Results paragraph one, how does Figure 2—figure supplement 5 relate to artemisinin spread. Further, what is "outcome" in this figure? It is not clear from the text or caption.

Outcome in this figure is proportional reduction in artemisinin resistance prevalence. Whilst is mentioned in the text, not having it in the legend was a glaring omission. Thank you for pointing this out.

– Results paragraph two, how is "a more static population" defined? Mobility is not well defined in the main text. A brief description is finally given in the second paragraph of the Discussion. Furthermore, mobility is described both as a fraction and as a decimal, which makes it difficult to compare across figures. How does connectivity relate to mobility? How does "well-connected" relate to mobility? Discussion paragraph four, does "low connectivity" mean low mobility?

We added a more detailed description on the definition of mobility values to Table 1 which would help the reader in interpreting the concept of mobility from early in the paper. Table 1 which is introduced to the reader at the beginning of the Results section gives an overall introduction to all the parameters varied in our simulations. We reworded “a more static population” to “a population of lower mobility” to avoid possible confusions. However, it should be clear that when individuals are highly mobile, villages become more connected, possibly sharing infection reservoirs. Surely, if there is no movement between two villages one could say they are not connected; and if there is a lot of population movement across two villages, those villages would be more connected. For consistency, we reworded these two lines to “…in well-connected populations (consisting of individuals with high mobility)…” and “…In low connectivity populations (consisting of individuals with low mobility) …” to avoid possible confusions to the reader.

– Results paragraph two, what constitutes "little difference"? Many of the curves in Figure 3—figure supplement 1 seem to be separated.

What we meant to convey was that even though the line in Figure 3 usually takes positive values early on, at year 10, it is either not very positive or visibly negative. In hindsight we acknowledge that it might have the opposite effect of what is intended and generate more confusion. We have removed that comparative argument from the sentence.

– In the same paragraph what are the "transmission heterogeneity distributions"? later in the paragraph "the other two" are mentioned but not named. They are not described anywhere clearly. Figure 4 is referenced here as a comparison between distributions but only uses a single distribution.

We added the word “transmission heterogeneity distribution” to Table 1 to help the reader locate the definition and pointed to the distributions plotted in Figure 1. “the other two” refers to “Normal and Log-Normal distributions” and this should now be clear in the main text. We reworded the sentence to “Figure 4 (Normal), Figure 4 supplement 1 (Log-Normal), and Figure 4—figure supplement 2 (Pareto)” to help direct the reader to the relevant figures.

– At the end of paragraph two, how is the "significant correlation" measured? There appear to be no statistical tests mentioned.

This was not rigorously quantified. We have changed the wording to “…there seems to be a correlation between…”

– Results paragraph three, how has vector control "greatly improved"? Is this measured? Visually they appear similar.

There seems to be some confusion here. The sentence states “…sustaining vector control for longer greatly improves…”. If we look at the figure mentioned at the end of that sentence, we see that the length of vector control period appears as the X-axis, and clearly has a major effect on the probability of elimination being reached. In some panels its effect goes as far as increasing the likelihood from 0 to over 90% for a fixed value of vector control efficacy.

– Discussion paragraph two, how are the mentioned factors a distant second and third? Where is this quantified?

This was quantified through univariate sensitivity analysis (see Author response image 1) which we chose not to present so as to not introduce too much entropy and focus primarily on the multivariate analysis in Figure 2. We have re-phrased this section of text to highlight that there are 3 variables the model is quite sensitive to, without quantifying the level of sensitivity. The sensitivity analyses already included as Figures 2—figure supplements 1-3 show how that sensitivity is manifested and how the different variables interact to produce an expected reduction in prevalence.

– How is a slower MDA implementation optimal? The solid line often appears to exceed the dashed line.

This is in direct juxtaposition with the statement in the previous sentence, where the conditions for a higher number of teams to be beneficial are presented. The solid line surpasses the dashed when the endpoint probability of elimination is high and more so for higher mobility levels. The only exceptions are simulations with 1MDA in which there are 2 seasonal peaks, which are addressed in the seasonality section.

More information is needed in the Materials and methods.– In paragraph seven mobility is briefly defined but referred to the main text, where there is not a good description. How are the nights chosen in the simulation? How does seasonal and long-term migration occur in the simulation? Does this effect the biting rate in places?

We added a more detailed description of mobility in Table 1 which is referred to the reader at the beginning of the Results section. Table 1 now also directs the reader to the Mobility section in the Simulation Protocol which explains how short-term movements are simulated. We also detail how the probability of an individual sleeping some place other than their house is evaluated on a daily basis, and its consequences for local transmission. Although different migration patterns are possible in our simulation platform, we clearly state that the scenarios contained here only include short-term movement of people. We should stress that the sensitivity of the model’s predictions to seasonal and long-term migration is much lower than to the general population short-term movement explored to great lengths throughout.

– In subsection “Simulation Protocol”, what other co-infecting parasites are considered?

The framework is extremely flexible so as to explore multi-species dynamics. However, here “co-infecting” means multiple infections with *Plasmodium falciparum* parasites, with drug-sensitive and drug-resistant as the categorical types.

– What drugs are used?

All treatments in the model use DHA-piperaquine PK/PD parameters. This is stated in the PK/PD methods section.

– Subsection “Model Initialisation”, do the data describe the initial properties of things like age, transmission status, etc.?

Yes, these properties are provided in the human input files. To clarify this to the reader, we have included an extensive section on model set-up in the Simulation Protocol which provides a clearer idea of how equilibrium conditions are assured for each model simulation presented in the manuscript.

– Subsection “Model Initialisation” paragraph two, two different implementations of how human agents are assigned is given, which is it?

We changed the text to: “Human agents are either randomly assigned a home village from a list of all available possible villages.”

– Subsection “Human Properties”, are cumulative number of exposures and immunity level set to 0 for humans of all ages? This seems unrealistic.

Apologies for the confusion. What we meant to say is that both properties have default values of 0 for newborns. We’ve corrected the text accordingly and have included an extensive section on model set-up in the Simulation Protocol which provides a clearer idea of how equilibrium conditions are assured for each model simulation presented in the manuscript.

– Subsection “Village Properties” is the calibration of biting rate stable when you introduce mobility and change malaria prevalence?

Indeed, changes in mobility imply a re-calibration of the biting rate. In our simulations, biting rates are calibrated for each combination of mobility, heterogeneity distribution, and prevalence to ensure system stability. We’ve added this clarification to the text.

– Subsection “Implementation of malaria relevant dynamics”, the implementation is unclear. A random number is drawn to see if there is an event and then random number are drawn to see if there are each of several events. Does this mean that an individual could be infected and treated and die all in the same day? If so, why go through the earlier event draw?

Yes, multiple events may happen to an individual on the same day. However, not all events are possible/valid to an individual given its status on a given day. For instance, a clinical resolution event or a treatment seeking is not valid to an individual until that individual develops clinical symptoms. For each individual, only events valid for its status are checked for occurrence. We’ve modified the text in this section accordingly. This is an outstanding question in these types of discrete time event simulation models, and we have been trying to develop optimisation methods to minimise the number of events evaluated per time step.

– Subsection “Human Population Dynamics”, when death happens, are individuals replaced by an infant or by an individual of any age? If the latter, how are their initial properties chosen.

An infant agent is created in place of the dead individual. We’ve added this description to the text.

– Subsection “Human Population Dynamics”, moi, cml, and lvl are not in Table 2. Could a table be created for parameters that change over time with an individual?

These are listed as properties of Human agents at the beginning of the Simulation Protocol. Parameters that change over time with an individual are given in that properties list.

– Can individuals be infectious immediately with program 1/sigma? This seems unrealistic. Why not use a fixed time delay?

Not right after a new infection has been acquired. Once an infection has entered an individual’s system, the intrinsic incubation process is scheduled to happen first. Once the infection has cleared the incubation process, its daily probability of becoming infectious is 1/*sigma*. However, only one of the three processes (1. acquisition of new infection, 2. incubation of the infection in the liver, 3. gametocytaemia) involved in the development of parasites in a human host may happen on a given day. A crucial part of the development and implementation process of the model was to benchmark the behaviour of each modular function, specifically by outputting the number of days it takes for specific human properties to change.

Fixed time delays can certainly be incorporated into our simulation platform. We conducted a brief comparison between the two methods and concluded that there are no significant qualitative differences in the results. We plan to integrate time delays to some of the model processes in future developments of our model so that a more formal comparison may be made.

– “Parasite killing rates depend on the person’s transmission status (s), with parasite clearance”, s is not in Table 2. Similar comment to above.

Transmission status is listed as a Human property. It is denoted *s* here for the convenience of the equation. We’ve added some texts to further explain its meaning and relation to variables (37-50) in Table 2.

– “Susceptibility/Infectiousness”, why include phic if the value is set to 1 (no increased/decreased relative infectiousness)?

Because its malleability is already implemented in the code. It is there so that this parameter can be informed by measures of gametocytaemia, rather than having a static number that is identical across all individuals. Unfortunately, we have not yet had the chance to calibrate this with an adequate dataset.

– Subsection “Mosquito Dynamics – Survival”, it appears these values are inverted.

Well spotted, thank you. We’ve corrected the text.

– "uncommon for any female to reach multiparity" Is this quantified?

We have quantified this effect by recording the number of concurrent infections in mosquitoes when they die. We changed the text to: “Their adult survivorship is quite limited though, making it uncommon for any female to reach multiparity with only 26% of female mosquitoes having more than one infection at the time of death. That number drops to ~14% for values of prevalence amenable to elimination.“

– How does immunity work? It's not discussed in detail, although some figures pertain to it.

“Immunity” is described as clinical immunity, i.e., protection against development of clinical symptoms, rather than a protection against infection. We calibrated a function describing how this immunity changes with exposure and an arbitrary immunity level which decreases over time and increases with recovery. This should now be a lot clearer after the changes made and additional details added to the Simulation Protocol.

Figures/Tables are hard to interpret due to missing information.– Figure 1: In A what are σ, ε, and ω? In C, what are the immunity levels? Is 1 higher or lower than 4?

Thanks for pointing these out. *σ* is the clinical probability, *ε* is the clinical symptom mellow rate, *ω* is the drug loss rate. We added more detailed description in Figure 1’s caption.

– Figure 2: x-axis is not labeled.

We added label to the revised Figure 2’s x-axis.

– Figure 3: Lots of things are varied but none are mentioned in the caption.

Parameters explored in Figure 3 are labelled on top of each column or to the sides of each row. We did not list individual parameters in the caption to avoid repetition. We have made changes to this figure’s labels and caption according to other reviewer comments which would hopefully help the readership.

– Figure 4: What is varied is not evident from the caption, nor is the distribution apparent.

Parameters that are varied in Figure 4 are labelled on top of each column and to the sides of each row. We did not list individual parameters in the caption to avoid repetition. We now mention the used distribution in the figure’s caption.

– Figure 1—figure supplement 1: Why cut off the distribution prior to 80 if that is the max?

Max age is the maximum allowed age in the model, not the maximum age recorded in the population. If there would be a person of that age in the model, their probability of dying would be such that they would be very likely to die soon. Figure 1—figure supplement 1 displays the equilibrium age profile, which in this case does not have individuals older than 70 years of age.

– Figure 1—figure supplement 2: What is "mean immunity level"? Why does it go up and down with age?

This figure has been replaced. More detail on how the clinical immunity function was fit to data might address these concerns. The immunity level can decrease if the person is not exposed for a long period of time, with probability 1*/alpha.*

– Figure 2—figure supplement 1: The size of the dots makes it hard to tell how far the lines go.– Figure 2—figure supplement 2: What is outcome here?– Figure 2—figure supplement 3: What is outcome here?– Figure 2—figure supplement 5: What is outcome here?

We’ve adjusted all of Figure 2’s figure supplements accordingly.

– Table 2: What does variable mean for parameters such as beta, previ, resit, vcefficacy? Can these be listed somewhere?

We added the following footnote to Table 2: “The values of these parameters are varied in different simulation settings. Their values are given in the description of each set of experiments.”

– Table 2: Parameter 37 – 50 are never mentioned in the manuscript.

These parameters are elements of the drug clearance rate matrix C introduced in the PK/PD section of the Simulation Protocol. We changed their notations in Table 2 and in the PK/PD section to hopefully make it clearer.

[Editors' note: further revisions were suggested prior to acceptance, as described below.]

Essential revisions:It is difficult to discern from the manuscript what things are functionally in the model and what are used in the analyses presented. These seem to be justified in the response to the reviewer but not clearly in the manuscript.

Without knowing what details of the model this is referring to, it is quite hard to address this point. We have gone back to the previous reply to reviewers and have made sure that everything included in those responses is also in the manuscript.

Attempt to capture realistic operational conditions, given that the paper is about the impact of actual operational conditions on MDA outcome. The MDA distribution structure and its implication on the authors' results are still not clear. The authors are adamant that they are not attempting to model an actual MDA distribution, but it seems that randomly selecting starting villages with which to seed MDA teams is unlikely to be how MDA is actually implemented, and potentially there are interactions between the spatial pattern of MDA implementation and the role of migration in MDA outcome. The authors must either show that their method of random selection of starting villages is how someone has operationalized MDA, or that a more operationally realistic village visitation order does not impact their findings.

Apart from targeted MDA strategies, where only a select few villages (selected according to some transmission metric) are to receive MDA, all other MDA strategies minimise the logistical entropy and will start in places where the logistical chains are more appropriate. This usually means radially from the control programmes’ headquarters. Since this is a synthetic population it seems appropriate to pick random locations. The assumption that a “strategic” MDA would obviously be better is baseless, since it is critically dependent on where the foci of transmission are. If they are heavily clustered over space then that could be something to consider, usually in a targeted MDA approach. In a full MDA approach where several foci can be scattered over space, random starting points should not be detrimental. Regardless, we have done thousands more simulations to unequivocally show that the operational MDA path is not a critical driver of success for areas in which foci can be scattered over space. This is now presented in Figure 3—figure supplement 2. Not only do we show that prioritizing high transmission villages for MDA is not better than selecting starting points at random, we also show that it is not superior to prioritizing low transmission villages. Note that if transmission is only occurring in a small area in space, a global MDA would never be an acceptable choice, and any control programme would instead opt for a targeted approach.

Infectivity: it is true that there is much that we do not understand quantitatively about infectivity/infectiousness. However, the simplifying assumption that all infected individuals, symptomatic and asymptomatic, have the same infectivity seems a little strong, especially since the authors note that changing phic does impact the effectiveness of MDA. Might it not also impact how the outcome of the MDA depends on the logistical, demographic, and transmission factors explored in this paper?

Essentially what the reviewer asked for here is a replica of 75% of all the work we had presented in the paper, with a different value for one of the parameters, which would take months. We opted to do the minimum set of simulations that we believe will satisfactorily resolve this issue and summarise their results in Figure 3—figure supplement 3. Consistently, and unlike models without explicit mosquito populations, when *phia* is low, there is an increased chance of elimination. After an extensive sensitivity analysis, we were able to determine this was solely due to the lower number of infectious sustaining transmission when *phia* is lower. To keep the mosquito population comparable to the simulations performed before, we leverage the biting rate in the associated *phia* simulations to obtain comparable prevalence levels. For a given prevalence value, we took the mosquito density in the *phia* =1 simulations and calibrated the biting rate (proxy for effective number of human/mosquito contacts) we would need when making *phia* = 0.2 to obtain the same prevalence at equilibrium. Since the infectiousness of asymptomatic infections is decreased, the biting rate will have to increase to reach the same level of prevalence. The result of this process is that the infection pool will be sustained in a smaller population of mosquitoes (which bite humans more frequently to make up for the decrease in human to mosquito infection efficiency). Thus, malaria elimination likelihood can be increased for *phia* = 0.2 due to a more likely crash of the population of infectious mosquitoes after MDA.